# Partial Identification with Proxy of Latent Confoundings via Sum-of-ratios Fractional Programming

**Zhiheng Zhang**[*1]

**Xinyan Su**[2,3]

[1]Institute for Interdisciplinary Information Sciences, Tsinghua University, Beijing, China
[2]Computer Network Information Center, Chinese Academy of Sciences, Beijing, China
[3]University of Chinese Academy of Sciences, Beijing, China

## Abstract

Causal effect estimation is a crucial theoretical tool in uncertainty analysis. The challenge of unobservable confoundings has raised concerns regarding quantitative causality computation. To address this issue, proxy control has become popular, employing auxiliary variables $W$ as proxies for the confounding variables $U$. However, proximal methods rely on strong assumptions, such as reversibility and completeness, that are challenging to interpret empirically and verify. Consequently, their applicability in real-world scenarios is limited, particularly when the proxies lack informativeness. In our paper, we have developed a novel optimization method named **P**artial **I**dentification with Proxy of Latent Confoundings via **S**um-of-Ratios **F**ractional **P**rogramming (PI-SFP). This method does not impose any additional restrictions upon proxies and only assumes the mild partial observability of the transition matrix $P(W \mid U)$. We have theoretically proven the global convergence of PI-SFP to the valid bound of the causal effect and analyzed the conditions under which the bounds could be tight. Our synthetic and real-world experiments validate our theoretical framework.

## 1 INTRODUCTION

Causal inference is crucial in uncertainty analysis across various fields such as medicine [Castro et al., 2020], economics [Hicks et al., 1980, Zhang et al., 2023b, 2020], and education [Peng and Knowles, 2003]. However, extracting useful causal information from observational data is challenging due to latent confoundings that can impede statistical association studies [Pearl, 2009a]. To address this

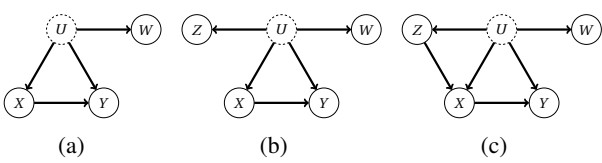

FIGURE 1: *Causal identification with confoundings via single-proxy control (a) or double-proxy control (b,c). $W$ or $Z$ are so-called confonuder proxies when identifying the causal effect of treatment $X$ on outcome $Y$.*

issue, researchers commonly rely on auxiliary variables for confounding adjustment. Representative auxiliaries include instrumental variables (IV) [Söderström and Stoica, 2002], proximal variables [Kuroki and Pearl, 2014, Tchetgen et al., 2020], or outcome-dependent variables [Gabriel et al., 2022].

In this paper, our primary focus is proximal causal identification, as it represents one of the most commonly utilized auxiliaries in our real world. The use of proxies for confounding adjustment has become a prominent topic of research both theoretically and empirically. Empirical work in this area dates back to [Wickens, 1972], which examined the potential benefits of proxies as an alternative to latent confounding in least square estimations. The concept has since been applied in observational studies such as Kolenikov and Angeles [2009], Wooldridge [2009] and further studied in other empirical works [Frost, 1979, Rothman et al., 2008]. On the other hand, theoretical research on this topic can be broadly categorized into two groups, which we illustrate in Fig.1: the "single-proxy scenario" (Figure1(a)) [Tchetgen et al., 2023, Park and Tchetgen, 2023] and the "double-proxy scenario" (Figures 1(b) and 1(c)) [Miao and Tchetgen, 2018, Shi et al., 2020, Singh, 2020, Kallus et al., 2021], where two confounding proxies $Z, W$ are available.

However, the application scope of proximal-based learning are still limited. For Figure 1(a), when both $W$ and $U$ are discrete random variables with a finite number of

[*]Correspondence author. Contacting Email: `zhiheng-20@mails.tsinghua.edu.cn`.

choices, Pearl [2012], Tchetgen et al. [2023], Park and Tchetgen [2023] proved the point-wise identifiability of the true causal quantities when the proxies are informative enough, e.g., the probability transition matrix $P(\boldsymbol{W} \mid \boldsymbol{U})$ is fully observable and reversible, which is the so-called completeness assumption. Stepping forward, when $P(\boldsymbol{W} \mid \boldsymbol{U})$ is not observable, Pearl extended the point-wise double-proxy cases (Figures 1(b), 1(c)) from Cai and Kuroki [2012], where both the exposure proxy control $\boldsymbol{Z}$ and the outcome proxy control $\boldsymbol{W}$ exist. Regretfully, so-called "double-negative control" [Miao et al., 2018, Cui et al., 2020, Tchetgen et al., 2020, Deaner, 2018, Shi et al., 2020, Singh, 2020, Nagasawa, 2018] methods and "single-proxy control" methods [Pearl, 2012, Tchetgen et al., 2023, Park and Tchetgen, 2023] are still subject to strict bridge functions, completeness assumptions [1], or their weaker forms Ghassami et al. [2023], Kallus et al. [2021].

These untestable and impractical constraints surrogates an important motivation: excessively strong conditions on proxy are imposed to sufficiently achieve the point-wise value of the causal effect, which would be violated in general cases. For instance, in a recommender system, an item's exposure to users is often considered as treatment $\boldsymbol{X}(\boldsymbol{X} = 0, 1)$, while the observed feedback is treated as outcome $\boldsymbol{Y}$. In this process, the user's socio-economic status and the item characteristics are used as latent confounders [Sato et al., 2020], which affect both $\boldsymbol{X}$ and $\boldsymbol{Y}$, and some observations of confounders are considered as proxy (e.g., item popularity ranking) $\boldsymbol{W}$ [Zhang et al., 2023a]. Unfortunately, the measurement of $\boldsymbol{W}$ and $\boldsymbol{U}$ would be inaccurate, and even worse, $P(\boldsymbol{W} \mid \boldsymbol{U})$ is irreversible due to the low dimension of $\boldsymbol{W}$, rendering the previous approach invalid. Hence, a natural scientific question arises: *How to conduct partial identification via partially observed proxies?*

To address this question, due to its originality and difficulty of double-proxies collection, we mainly focus on the single proxy case (Fig.1(a)), which is mostly related to Pearl [2012], Tchetgen et al. [2023], Park and Tchetgen [2023], and our method could be naturally generalized to Fig.1(b)-Fig.1(c). We weaken the requirement of "total precise observability" of $P(\boldsymbol{W} \mid \boldsymbol{U})$ to "partial observability", and thus generalize the point-wise identification into a partial identification. Our method contributes to the traditional fractional programming methods [Stancu-Minasian, 2012] since we do not rely on strong concavity assumption. More importantly, our method also advances the state-of-the-art constrained-optimization-based literature upon partial identification Duarte et al. [2023], Li and Pearl [2022], since we additionally provide a non-parametric convergence rate to causal queries based on branch-and-bound strategy. Our

contributions are summarized as follows:

- We generalize the traditional proximal learning literature from point-wise identification to partial identification, with a more reasonable and weaker partial observability assumption.

- We introduce a global optimization strategy called PI-SFP and theoretically prove that it can globally converge to the valid bound of casual queries. Moreover, we justify whether the bound is tight. Synthetic and real-world experiments have demonstrated our findings.

- We theoretically justify the necessity of the partial observability assumption in proximal control. It is supported by the negative result that traditional informative proxies might not sufficiently guarantee informative partial identification.

## 2 PRELIMINARIES AND FRAMEWORK

The concept of causal effect is closely linked with the 'do' operator, which can be viewed as an external intervention [Pearl et al., 2000, Pearl, 2009b]. In particular, the causal effect of treatment $\boldsymbol{X}$ on outcome $\boldsymbol{Y}$ is represented as $f(y \mid do(x))$ in Fig.1, where $do(x)$ signifies that the treatment $\boldsymbol{X}$ is fixed at a specific value $x$, and $f(\cdot)$ denotes the probability mass/density function for discrete/continuous variables. We use $d_u$ to denote the cardinality of confounders. As per the back-door criteria [Pearl et al., 2000], the identification of $f(y \mid do(x))$ is given by $\sum_{i=1}^{d_u} f(y \mid u_i, x) f(u_i)$, namely $f(y, x) + \sum_{i=1}^{d_u} f(y, u_i, x) f(u_i, \neg x) / f(u_i, x)$.

Such decomposition results from $f(u_i) = f(u_i, x) + f(u_i, \neg x)$. In Kuroki and Pearl [2014], the authors assumed that the transition matrix $P(\boldsymbol{W} \mid \boldsymbol{U})$ is observable and reversible, and hence claimed that $f(y \mid do(x))$ is identifiable. In other words, the value of each item as above can be explicitly extracted as follows[2]:

$$\begin{bmatrix} f(y, \boldsymbol{U}, x) \\ f(\boldsymbol{U}, x) \\ f(\boldsymbol{U}, \neg x) \end{bmatrix} = P(\boldsymbol{W} \mid \boldsymbol{U})^{-1} \begin{bmatrix} f(y, \boldsymbol{W}, x) \\ f(\boldsymbol{W}, x) \\ f(\boldsymbol{W}, \neg x) \end{bmatrix}. \quad (1)$$

Our paper focuses on generalization, where we consider the partial identification of $f(y \mid do(x))$ instead of its unique form computation. This change stems from our relaxed assumption on $P(\boldsymbol{W} \mid \boldsymbol{U})$, where we move from total observability to partial observability and remove the guarantee

---

[1]The previous reversibility assumption of $P(\boldsymbol{W} \mid \boldsymbol{U})$ was strengthened to that of $P(\boldsymbol{Z}, \boldsymbol{W} \mid x)$ and $P(y, \boldsymbol{Z}, \boldsymbol{W} \mid x)$. Moreover, the path $\boldsymbol{W}$ to $\boldsymbol{Y}$ in Fig. 1(c) can be additionally permitted.

[2]In our paper, we use bold letters to denote column vectors of corresponding possible values. For instance, $f(y, \boldsymbol{U}, x) = [f(y, u_1, x), f(y, u_2, x), ... f(y, u_{d_u}, x)]^T$. Furthermore, if a symbol has two bold letters such as $P(\boldsymbol{W} \mid \boldsymbol{U})$, it denotes the matrix $[f(\boldsymbol{W} \mid u_1), f(\boldsymbol{W} \mid u_2), ... f(\boldsymbol{W} \mid u_{d_u})]$, where $f(\boldsymbol{W} \mid u_i) = [f(w_1 \mid u_i), f(w_2 \mid u_i), ... f(w_{d_w} \mid u_i)]^T, i = 1, 2, ... d_u$.

for reversibility (thus $P(\boldsymbol{W} \mid \boldsymbol{U})^{-1}$ in Eqn (2) may not exist). Specifically, we expand the identification region of $P(\boldsymbol{W} \mid \boldsymbol{U})$ from a fixed distribution to the family $\mathscr{P}$, which contains all possible $P(\boldsymbol{W} \mid \boldsymbol{U})$ such that $P(\boldsymbol{W} \mid \boldsymbol{U}) - \underline{P(\boldsymbol{W} \mid \boldsymbol{U})}$ and $\overline{P(\boldsymbol{W} \mid \boldsymbol{U})} - P(\boldsymbol{W} \mid \boldsymbol{U})$ all both non-negative. Here $\underline{P(\boldsymbol{W} \mid \boldsymbol{U})}$ and $\overline{P(\boldsymbol{W} \mid \boldsymbol{U})}$ are two priori known matrices to bound $P(\boldsymbol{W} \mid \boldsymbol{U})$. This scenario is prevalent in the real world. While studies [Kuroki and Pearl, 2014, Greenland, 2005] have generally confirmed its verifiability, recent literature has not fully explored it. In our paper, we reiterate the condition $P(\boldsymbol{W} \mid \boldsymbol{U}) \in \mathscr{P}$ as the 'partial observability assumption' in our following text. Under this assumption, we can naturally set our original goal as seeking the lower bound of $f(y \mid do(x))$ (upper bound is symmetric) by solving the following partial identification problem: $f(y,x) + \min \sum_{i=1}^{d_u} f(y,u_i,x) f(u_i, \neg x)/f(u_i,x)$, subject to $f(y, \boldsymbol{W}, \boldsymbol{U}, \boldsymbol{X}) \in \mathcal{F}$. Here $f(y, \boldsymbol{W}, \boldsymbol{U}, \boldsymbol{X})$ is a three-order $d_w * d_u * d_x$ tensor indicating the joint probability distribution of each $w \in \boldsymbol{W}, u \in \boldsymbol{U}, x \in \boldsymbol{X}$ together with $\boldsymbol{Y} = y$. The set $\mathcal{F} = \{f(y, \boldsymbol{W}, \boldsymbol{U}, \boldsymbol{X}) : f(y, \boldsymbol{W}, \boldsymbol{U}, \boldsymbol{X})$ is compatible with $P(\boldsymbol{W} \mid \boldsymbol{U}) \in \mathscr{P}\}$.

Achieving this goal is challenging due to the difficulty in achieving its tight bound. Firstly, the feasible region $\mathcal{F}$ is challenging to represent in a closed form due to the boundary constraints that include the partially observable $P(\boldsymbol{W} \mid \boldsymbol{U})$. This constraint can be seen as a first-kind Fredholm integral equation, which is an ill-posed problem when $P(\boldsymbol{W} \mid \boldsymbol{U})$ is irreversible[3]. To address this issue, we propose relaxing the feasible region from $\mathcal{F}$ to $\widetilde{\mathcal{F}}$ ($\mathcal{F} \subseteq \widetilde{\mathcal{F}}$), which contains a closed-form expression. Specifically, the relaxed condition $f(y, \boldsymbol{W}, \boldsymbol{U}, \boldsymbol{X}) \in \widetilde{\mathcal{F}}$ ensures that the feasible regions of $f(y, \boldsymbol{U}, x)$, $f(y, \boldsymbol{U}, x)$, and $f(\boldsymbol{U}, \neg x)$ can be represented in a calculable closed-form. We refer to these as $IR_{F(y,U,x)}$, $IR_{F(U,x)}$, and $IR_{F(U,\neg x)}$, respectively, in our final objective function in the following section.

Secondly, even if we retreat and seek its valid bound as above, it remains non-trivial due to the difficulty in finding an appropriate optimization method. Since the causal effect is expressed as a fractional summation, it is natural to explore techniques in sum-of-ratios fractional programming (SFP). The general form of SFP, as summarized in [Schaible and Shi, 2003], is represented as follows: $\min\{\sum_{i=1}^{M} \frac{g_{1i}(\phi)}{g_{2i}(\phi)}\}, \phi \in S\}, g_{1i}(\phi)$ is convex, $g_{2i}(\phi)$ is concave, $g_{1i}(\phi), g_{2i}(\phi) > 0$. Here $S$ is a convex set, and $M \geq 2$ is a integer. In order to ensure the global nature of optimal solutions, $g_{1i}(\Phi)$ and $g_{2i}(\Phi)$ are assumed to be convex and concave, respectively. In contrast with the above formulation, we should choose $\phi = ((f(y, u_i, x), ...)^T, (f(u_i, x), ...)^T, (f(u_i, \neg x), ...)^T)$, and $g_{1i}(\phi) = f(y, u_i, x) f(u_i, \neg x), g_{2i}(\phi) = f(u_i, x)$. More-

over, $M = d_u, i = 1, 2, ...d_u$. However, this construction violates the traditional convex-concave assumption, as $g_{1i}(\phi)$ is not convex. Therefore, here traditional SFP algorithms [Schaible and Shi, 2003] are not suitable.

To address these two challenges, we introduce the Partial Identification with Sum-of-Ratios Fractional Programming (PI-SFP) algorithm. Motivated by the branch and bound strategy [Dai et al., 2005, Lawler and Wood, 1966, Dur et al., 2001, Pei and Zhu, 2013] and DC programming [Horst and Thoai, 1999, Tao and An, 1997, Pei and Zhu, 2013], our algorithm iteratively searches the optimal bound through feasible region partitioning. Different from the closely-related partial identification literature Duarte et al. [2023], Li and Pearl [2022], we also provide a comprehensive new convergence analysis.

For supplement, due to these two challenges of solving the partial observability case, another line of recent literature has avoided further discussion on the observability of $P(\boldsymbol{W} \mid \boldsymbol{U})$. Instead, researchers have introduced an auxiliary variable $\boldsymbol{Z}$ and formalized the problem as the double negative control [Miao et al., 2018, Cui et al., 2020, Tchetgen et al., 2020, Deaner, 2018, Shi et al., 2020, Singh, 2020, Nagasawa, 2018, Kallus et al., 2021]. However, as illustrated in the introduction, there are no free lunch (Table. 2). These works are restricted by additional assumptions about proxies, such as the completeness condition and the bridge function condition. Importantly, all of these works still rely on the reversibility of $P(\boldsymbol{W} \mid \boldsymbol{U})$, except for Ghassami et al. [2023], Kallus et al. [2021], who substituted it as a weaker bridge function condition. Consequently, even if the transition matrix is still reversible, but with a large conditional number[4], numerical computations would already become extensively overwhelming. In conclusion, revisiting single-proxy control under the partial observability of $P(\boldsymbol{W} \mid \boldsymbol{U})$ is not only challenging but also necessary.

## 3 METHOD

We begin by presenting definitions and assumptions and then rigorously formulate the objective function.

**DEFINITIONS** We denote $Y, Y_0, Y_1 \in [Y^L, Y^U], Z \in [Z^L, Z^U], X \in [X^L, X^U], W \in [W^L, W^U], U \in [U^L, U^U]$. Moreover, we use $d_z, d_u, d_w, d_x$ to denote the cardinality of variables $\boldsymbol{Z}, \boldsymbol{U}, \boldsymbol{W}, \boldsymbol{X}$. For instance, the set of confounder $\boldsymbol{U}$ is $\{u_1, u_2, ...u_{d_u}\}$. $X \neq x$ is simplified as $\neg x$. $Y$ can be discrete or continuous.

Moreover, $Y_x$ is the value of $\boldsymbol{Y}$ when $X$ is forced to be $x$.

---

[3]Otherwise, the closed-form expression of $\mathcal{F}$ can only be approximated iteratively by complex numerical methods [Strand and Westwater, 1968], which is beyond our scope.

[4]The conditional number of matrix $A$ is denoted as $\kappa(A) = \frac{\sigma_{max}(A)}{\sigma_{min}(A)}$, where $\sigma_{max}(A)$ and $\sigma_{min}(A)$ denote the maximal and minimal singular values of $A$. If some rows/columns of $A$ are similar (or equal), then $\kappa(A)$ is large (or $+\infty$), and $A^{-1}$ is computationally hard (or may not exist).

$\pi(x)$ is a weight function[5] of $\boldsymbol{X}$. On this basis, $ACE_{\boldsymbol{X} \to \boldsymbol{Y}}$ denotes the average causal effect (ACE) from $\boldsymbol{X}$ to $\boldsymbol{Y}$, namely that $ACE_{\boldsymbol{X} \to \boldsymbol{Y}} = \int_x \int_y y f(Y_x = y)\pi(x)dxdy$.

**Assumption 1** *(partial observability)* $P(\boldsymbol{W} \mid \boldsymbol{U}) \in \mathscr{P}$.

Here the set $\mathscr{P}$ is identified in Section 2, where $\underline{P(\boldsymbol{W} \mid \boldsymbol{U})}$ and $\overline{P(\boldsymbol{W} \mid \boldsymbol{U})}$ are two a priori known matrices serving as the partial order of $P(\boldsymbol{W} \mid \boldsymbol{U})$.

**OBJECTIVE FUNCTION**  The objective of this section is to formalize the optimization problem of the single proxy control under Assumption. 1. During this process, we aim to tackle the two challenges introduced in the preliminaries. Our primary objective is (the maximum case is symmetric):

$$\min f(y,x) + \sum_{i=1}^d \frac{f(y,u_i,x)f(u_i,\neg x)}{f(u_i,x)}, \qquad (2)$$
$$\text{subject to: } f(y,\boldsymbol{U},\boldsymbol{W},\boldsymbol{X}) \in \mathcal{F}.$$

Here $\mathcal{F} = \{f(y,\boldsymbol{U},\boldsymbol{W},\boldsymbol{X}) : f(y,\boldsymbol{U},\boldsymbol{W},\boldsymbol{X}) \text{ is compatible with Assumption. 1 and observed } f(y,\boldsymbol{W},\boldsymbol{X})\}$.

As we suggested in the preliminaries, *the first challenge* is the nonexistence of closed-form expression of $\mathcal{F}$. To solve it, we introduce the new symbol $\widetilde{\mathcal{F}}$ to formally describe the relaxation of the identification region of $f(y,\boldsymbol{U},\boldsymbol{W},\boldsymbol{X})$. For preparation, we introduce the symbol $\boldsymbol{\theta}, \boldsymbol{\psi}, \boldsymbol{\omega}$ and follow the previous notation $\boldsymbol{\phi}$: $\theta_i = f(y,u_i,x)$, $\psi_i = f(u_i,x)$, $\omega_i = f(u_i,\neg x)$, $\boldsymbol{\theta} = (\theta_1,\theta_2,...\theta_d)^T$, $\boldsymbol{\psi} = (\psi_1,\psi_2,...\psi_d)^T$, $\boldsymbol{\omega} = (\omega_1,\omega_2,...\omega_d)^T$, $\boldsymbol{\phi} = (\boldsymbol{\theta}\,\boldsymbol{\psi}\,\boldsymbol{\omega})$.

Then we construct a broader set $\widetilde{\mathcal{F}}$ as follows:

$$\widetilde{\mathcal{F}} = \left\{ f(y,\boldsymbol{U},\boldsymbol{W},\boldsymbol{X}) : \boldsymbol{\phi} \in IR_{\boldsymbol{\Phi}}, IR_{\boldsymbol{\Phi}} = IR_{\boldsymbol{\Phi}}^1 \cap IR_{\boldsymbol{\Phi}}^2 \right\},$$

where the set $IR_{\boldsymbol{\Phi}}^1$ denotes the set of $\boldsymbol{\Phi}$ satisfies $\overline{P(\boldsymbol{W} \mid \boldsymbol{U})}\boldsymbol{\phi} \geq f(y,\boldsymbol{W},x), f(\boldsymbol{W},x), f(\boldsymbol{W},\neg x) \geq \underline{P(\boldsymbol{W} \mid \boldsymbol{U})}\boldsymbol{\phi}$. Here we use $\boldsymbol{S}_1 \geq \boldsymbol{S}_2$ to denote $\boldsymbol{S}_1 - \boldsymbol{S}_2$ is a non-negative matrix, and $\boldsymbol{I_{d*d}}$ denotes the $d*d$ identity matrix. Moreover, the set $IR_{\boldsymbol{\Phi}}^2$ indicates all possible $\boldsymbol{\phi}$ such that $\boldsymbol{1} * \boldsymbol{\theta} = f(y,x), \boldsymbol{1} * \boldsymbol{\phi} = f(x), \boldsymbol{1} * \boldsymbol{\omega} = f(\neg x)$, and $\forall i, \theta_i \in [0, f(y,x)], \phi_i \in (0, f(x)], \omega_i \in [0, f(\neg x)]$.

Here $\boldsymbol{1}$ denotes the corresponding all-ones vector. By this construction, the enclosure property $\mathcal{F} \subseteq \widetilde{\mathcal{F}}$ is guaranteed.

**Proposition 1** $\mathcal{F}$ *is enclosed by* $\widetilde{\mathcal{F}}$, *namely that* $\mathcal{F} \subseteq \widetilde{\mathcal{F}}$.

The proof is shown in the Appendix A.1. Proposition. (1) provides the extension of the feasible region

[5] In [Kallus et al., 2021], it is called as generalized average causal effect. It can degenerate to the traditional form [Pearl, 2013] as $ACE_{\boldsymbol{X} \to \boldsymbol{Y}} = E(Y_1) - E(Y_0)$ if we choose $d_x = 2, X = \{0,1\}$, and $\pi(x) = \boldsymbol{sgn}(x)$, where $\boldsymbol{sgn}(\cdot)$ is the sign function.

of $f(y,\boldsymbol{U},\boldsymbol{W},\boldsymbol{X})$ from $\mathcal{F}$ to $\widetilde{\mathcal{F}}$. On this basis, Eqn (2) is relaxed as follows: $\underline{f(Y_x = y)} = \min f(y,x) + \sum_{i=1}^d \theta_i\omega_i/\psi_i$, subject to: $f(y,\boldsymbol{U},\boldsymbol{W},\boldsymbol{X}) \in \widetilde{\mathcal{F}}, i.e., \boldsymbol{\phi} \in IR_{\boldsymbol{\Phi}}$.

Symmetrically, the optimal value is denoted as $\overline{f(Y_x = y)}$ for the maximum case. Moreover, the corresponding set of optimal solutions are denoted as $\boldsymbol{\Phi_{opt}}$. The following proposition discuss the tightness of $\underline{f(Y_x = y)}$:

**Proposition 2** *The outcome* $\underline{f(Y_x = y)}$ *serves as the valid lower bound of* $f(Y_x = y)$. *Moreover, this bound is tight if and only if the following set is not empty:* $\Big\{ f(y,\boldsymbol{U},\boldsymbol{W},\boldsymbol{X}) : f(y,\boldsymbol{U},\boldsymbol{W},\boldsymbol{X}) \in \mathcal{F}$ *and is compatible with some* $\boldsymbol{\phi_{opt}} \in \boldsymbol{\Phi_{opt}} \Big\} \neq \emptyset$, *where* $\boldsymbol{\phi_{opt}}$ *is an element of the set* $\boldsymbol{\Phi_{opt}}$. *The maximum case* $\overline{f(Y_x = y)}$ *is symmetric.*

As discussed in the preliminaries, we have explained the reasons why guaranteeing that $\underline{f(Y_x = y)}$ is a tight bound is beyond the community scope, hence Proposition 2 is already the optimal result. In practice, it could be verified that Eqn (2) holds in a number of cases, whose details are detailed in Appendix A.2.

We now aim to tackle *the second challenge*: the non-trivial nature of the fractional programming problem due to the invalidation of the convex-concave condition. To address this issue, we adopt the *difference-in-convex (DC)* decomposition strategy to formally describe how we relax the above formulation into a linear programming problem. To prepare for this, we first transform the fractional form and introduce the knockoff variable $\boldsymbol{\psi^o}$ to replace the denominator. Next, we introduce the $4d-$ dimensional vector $\boldsymbol{\gamma}$: $\boldsymbol{\psi^o} = (\psi_1^o, \psi_2^o, ...\psi_d)^T$, $\boldsymbol{\gamma} = \left((\boldsymbol{\psi^o})^T, \boldsymbol{\theta}^T, \boldsymbol{\psi}^T, \boldsymbol{\omega}^T\right)^T$, where $(\boldsymbol{\theta}, \boldsymbol{\psi}, \boldsymbol{\omega})$ is copied from $\boldsymbol{\phi}$.

Then our original function is equivalently transformed to

$$\underline{f(Y_x = y)} = \min f(y,x) + \sum_{i=1}^d \psi_i^o \theta_i \omega_i \qquad (3)$$
$$\text{subject to : } \boldsymbol{\gamma} \in IR_{\Gamma} := \{\boldsymbol{\gamma} : \boldsymbol{\phi} \in IR_{\boldsymbol{\Phi}}, \psi_i^o \psi_i = 1\}.$$

Here $i = 1,...d$. Although the knock-off trick has been implemented, achieving the final goal remains challenging in practice. Firstly, the objective function and constraints are both non-convex and nonlinear. Secondly, relying on local optimal algorithms alone is not viable, as it may not guarantee the validity of the bound $\underline{f(Y_x = y)}$. Thus, our motivation is to construct a weaker linear programming form that can approximate the global optimal value of (3). To achieve this, we propose applying the *difference-in-convex (DC)* decomposition as our core idea: $\forall \boldsymbol{\gamma}$,

$$\sum_{i=1}^d \psi_i^o \theta_i \omega_i = C_1(\boldsymbol{\gamma}) - C_2(\boldsymbol{\gamma}), \quad \psi_i^o \psi_i = D_{i1}(\boldsymbol{\gamma}) - D_{i2}(\boldsymbol{\gamma}),$$
$$(4)$$

where $i = 1, 2, \cdots, d, C_1(\boldsymbol{\gamma}), C_2(\boldsymbol{\gamma}), D_{i1}(\boldsymbol{\gamma}), D_{i2}(\boldsymbol{\gamma})$ [6] are all convex functions (see Appendix A.5) satisfying that $C_1(\boldsymbol{\gamma}), C_2(\boldsymbol{\gamma}), D_{i1}(\boldsymbol{\gamma}) :=$

$$\sum_{i=1}^{d} \frac{1}{6} (\sum_{cyc} \psi_i^o)^3 + \frac{1}{2} \sum_{cyc} (\psi_i^o)^4 + \frac{1}{2} \sum_{cyc} (\psi_i^o)^2,$$

$$\sum_{i=1}^{d} \frac{1}{6} \sum_{cyc} (\psi_i^o)^3 + \frac{1}{4} \sum_{cyc} [(\psi_i^o)^2 + \theta_i]^2 + [\psi_i^o + \theta_i^2]^2, \quad (5)$$

$$\frac{1}{2} (\psi_i^o + \psi_i)^2, \; D_{i2}(\boldsymbol{\gamma}) = \frac{1}{2} [(\psi_i^o)^2 + (\psi_i)^2],$$

respectively. Exploiting their convexity, we bound them by the following linear functions, which are constructed by secants and tangents of the original function:

$$C_1(\boldsymbol{\gamma}) - C_2(\boldsymbol{\gamma}) \geq C_1^{\tan}(\boldsymbol{\gamma}) - C_2^{\sec}(\boldsymbol{\gamma}),$$
$$D_{i1}(\boldsymbol{\gamma}) - D_{i2}(\boldsymbol{\gamma}) \in [D_{i1}^{\tan}(\boldsymbol{\gamma}) - D_{i2}^{\sec}(\boldsymbol{\gamma}), D_{i1}^{\sec}(\boldsymbol{\gamma}) - D_{i2}^{\tan}(\boldsymbol{\gamma})]$$
$$(6)$$

For their explicit form solutions, we refer the readers to (10). This allows us to relax the original problem in (3) into the following linear program:

$$\min f(y, x) + C_1^{\tan}(\boldsymbol{\gamma}) - C_2^{\sec}(\boldsymbol{\gamma})$$
$$\text{subject to} : \boldsymbol{\phi} \in IR_\Phi, D_{i1}^{\tan}(\boldsymbol{\gamma}) - D_{i2}^{\sec}(\boldsymbol{\gamma}) \leq 1, \quad (7)$$
$$D_{i1}^{\sec}(\boldsymbol{\gamma}) - D_{i2}^{\tan}(\boldsymbol{\gamma}) \geq 1, i = 1, 2, \cdots, d.$$

It is clear that this shift causes the estimation error. In order to eliminate it in practice, we iteratively do DC within simplicial partitioned feasible regions. Details will be shown in the following section. In conclusion, we already address these two challenges in the preliminaries. In addition, Our framework is natural to extend to the ACE cases, and we refer readers to Appendix A.4 for details.

## 4 ALGORITHM

In this section, we demonstrate how to compute $\underline{f(Y_x = y)}$ in (3) practically. As mentioned earlier, this involves optimizing a non-convex function, which requires new optimization techniques to find the global optimum. To this end, we propose Partial Identification via Sum-of-ratios Fractional Programming (PI-SFP), a fractional programming-based method that optimizes the objective through iterative approximation. Specifically, we begin by constructing a simplex $S_0$ that encloses the feasible region of (3). We then use $S_0$ to identify a lower bound of $\underline{f(Y_x = y)}$ using the difference-in-convex (DC) decomposition strategy. In each iteration, we partition $S_0$ into multiple simplices to refine the lower bound constructed in the initial step.

---

[6]Note that the sub-script $cyc$ in (5) is an abbreviation of cyclic sum following [Du et al., 2012], which cycles through $\{\psi_i^o, \theta_i, \omega_i\}$ in the corresponding function and take the sum. For instance, we have $\sum_{cyc} [\psi_i^o + \theta_i^2]^2 = [\psi_i^o + \theta_i^2]^2 + [\theta_i + \omega_i^2]^2 + [\omega_i + (\psi_i^o)^2]^2.$

---

**Algorithm 1:** Partial Identification via Sum-of-ratios Fractional Programming (PI-SFP).

**Input:** Observational distribution $f(y, \boldsymbol{W}, \boldsymbol{X})$, $\underline{P(\boldsymbol{W} \mid \boldsymbol{U})}, \overline{P(\boldsymbol{W} \mid \boldsymbol{U})}$, a prespecified error bound $\delta > 0$.

**Output:** A lower bound estimate $\underline{f_{opt}^k(Y_x = y)}$.

1 Let $k = 0$, construct an original simplex $S_0 = $ **Initialization**$(f(y, \boldsymbol{W}, \boldsymbol{X}), \underline{P(\boldsymbol{W} \mid \boldsymbol{U})}, \overline{P(\boldsymbol{W} \mid \boldsymbol{U})})$;

2 Calculate a lower bound of $f_{S_0}(Y_x = y)$ via the **Bounding** function: $\underline{f_{S_0}(Y_x = y)} = $ **Bounding**$(S_0)$;

3 Set the collection of simplices at the 0-th iteration as $\mathcal{S}_0 = \{S_0\}$;

4 **while** *PI-SFP$_{error}$* $\leq \delta$ **do**

5     Let $\tilde{S}_k = \text{argmin}_{S \in \mathcal{S}_k} \underline{f_S(Y_x = y)}$, where $\underline{f_S(Y_x = y)}$ denotes the output of **Bounding**$(S)$ with input $S$;

6     Split $\tilde{S}_k$ into two simplices $\tilde{S}_{k1}$ and $\tilde{S}_{k2}$ via the **Bisection** function: $\tilde{S}_{k1}, \tilde{S}_{k2} = $ **Bisection**$(\tilde{S}_k)$ and set $\mathcal{S}_{k+1} = (\mathcal{S}_k \setminus \tilde{S}_k) \cup \{\tilde{S}_{k1}, \tilde{S}_{k2}\}$;

7     Calculate the estimation error bound via *PI-SFP$_{error}$* $= $ **Global_error**$(\tilde{S}_0, \tilde{S}_1, \cdots, \tilde{S}_{k+1})$;

8     Set $k = k + 1$;

9 Return $\underline{f_{opt}^k(Y_x = y)} = \max_{i \in \{0, 1, \dots k\}} \underline{f_{\tilde{S}_i}(Y_x = y)}$.

---

The remainder of this section is organized as follows. In Section. 4.1, we introduce the main framework of our algorithm, which we divided into four modules: 1) **Initialization**, 2) **Bisection**, 3) **Bounding**, and 4) **Global_error**. We then elaborate on each module in detail in Section. 4.2. For ease of notation, we introduce the following symbols for algorithm description: • For a simplex $S$, $dia(S) := \max_{s_1, s_2 \in S} \|s_1 - s_2\|_2$ denotes its diameter, and $S^i$ denotes its $i$−th supporting vector, $i = 0, 1, \dots 4d$. • $f_S(Y_x = y)$ denotes the optimal value of (3) when its feasible region is strengthened to $\boldsymbol{\gamma} \in IR_\Gamma \cap S$.

### 4.1 OVERVIEW OF PI-SFP

The framework of PI-SFP to solve (3) is as follows.

**Step 1** involves pre-processing, where the **Initialization** function is used to construct a baseline simplex $S_0$ enclosing the original feasible region, such that $IR_\Gamma \subseteq S_0$ and $\underline{f(Y_x = y)} = \underline{f_{S_0}(Y_x = y)}$ (see Lemma 2 in Appendix A.5). This equivalent transformation enables the computation of $\underline{f(Y_x = y)}$ via $\underline{f_{S_0}(Y_x = y)}$. In **Step 2**, we use the DC decomposition strategy as in (4)-(7) to find a lower bound of $\underline{f_{S_0}(Y_x = y)}$, which is denoted by $\underline{f_{S_0}(Y_x = y)}$.

In **Steps 4-9**, we employ a bisection-like approach to iteratively partition $S_0$ into a set of simplices $\mathcal{S}_k$ in the $k$-th

**Algorithm 2:** Recursive procedure to split the simplicial partitions (**Bisection**).

---

**Input:** Simplex $S$ with vertices $\{S^0, \cdots, S^{4d}\}$.
**Output:** Two new simplices $S_1, S_2$.

1 Set $S^{t_1}, S^{t_2}$ as the vertices incident to the longest edge of $S$: $\{t_1, t_2\} = \underset{\{a,b\}\in\{0,1,\cdots,4d\}}{\arg\max} \|S^a - S^b\|_2$;

2 Construct $S_1, S_2$ based on the following two sets of vertices: $\{S^0, \cdots, S^{t_1-1}, v, S^{t_1+1}, \cdots, S^{4d}\}$, $\{S^0, \cdots, S^{t_2-1}, v, S^{t_2+1}, \cdots, S^{4d}\}$, where $v$ corresponds to the midpoint the longest edge.

---

iteration. Next, we reapply the DC decomposition strategy to new simplices to obtain a more accurate estimate. We stop and return the lower bound estimate in **Step 10** once the bounding error calculated by **Global_error** reaches the prespecified threshold $\delta$. Otherwise, we make more delicate partitions and iterate this step.

## 4.2 IMPLEMENTATION OF PI-SFP

In this section, the above four functions are illustrated in detail. **1) Initialization()**: The objective of this function is to construct an original simplex $S_0$ that encloses the feasible region $IR_\Gamma$. To achieve this, we draw inspiration from [Horst et al., 2000, Pei and Zhu, 2013] and use the following approach to construct $S_0$. The justification of such construction is given in lemma. (2).Here $S_0$ denotes the set of $\gamma = (\gamma_1, \cdots \gamma_{4d})$ satisfying $1 \leq i \leq 4d_u, \mathbf{1} * \gamma \leq \alpha, \gamma_i \geq \gamma_i^l := \min_{\gamma\in IR_\Gamma} \gamma_i$, and $\alpha = 1 + f(y,x) + \frac{d^2(\psi^l+\psi^u)^2}{4f(x)\psi^l\psi^u}$, where

$$\psi^l = \min_{i\in[2d+1,3d]} \gamma_i^l, \psi^u = \min_{i\in[2d+1,3d]} \gamma_i^u, \gamma_i^u := \max_{\gamma\in IR_\Gamma} \gamma. \quad (8)$$

**2) Bisection()**: Motivated by the approach proposed in Rivara [1984], the goal of this function is to partition an input simplex $S$ into two simplices $S_1$ and $S_2$ using the longest-edge (LE) bisection strategy. The partitioning details are outlined in Algorithm 2.

**3) Bounding()**: The purpose of this function is to derive a lower bound of $\underline{f_S(Y_x = y)}$ using the input $S$. This is the most crucial element of the algorithm. It is worth remembering that $\underline{f_S(Y_x = y)}$ can be expressed as the solution of the optimization program (3) with an additional constraint $\gamma \in S$. With the derivations shown in (4)-(7), we can easily obtain a lower bound of $\underline{f_S(Y_x = y)}$ by solving the follow-

ing optimization problem[7]:

$$\underline{f_S(Y_x = y)} = \min f(y,x) + C_1^{\tan}(\gamma) - C_2^{\text{sec}}(\gamma)$$

$$\text{subject to: } \phi \in IR_\Phi, \gamma \in S; D_{i1}^{\tan}(\gamma) - D_{i2}^{\text{sec}}(\gamma) \leq 1,$$
$$D_{i1}^{\text{sec}}(\gamma) - D_{i2}^{\tan}(\gamma) \geq 1, i = 1, ...d.$$
$$(9)$$

As demonstrated in lemma. (5) of Appendix A.5, the functions $C_1^{\tan}(\gamma)$, $C_2^{\text{sec}}(\gamma)$, $D_{i1}^{\tan}(\gamma)$, $D_{i2}^{\text{sec}}(\gamma)$, $D_{i1}^{\text{sec}}(\gamma)$ and $D_{i2}^{\tan}(\gamma)$ are constructed from $C_1(\gamma), C_2(\gamma)$, $D_{i1}(\gamma), D_{i2}(\gamma)$'s in (5) based on **sec**ants and **tan**gents within the simplex $S$:

$$\begin{bmatrix} C_k^{\tan}(\gamma) \\ D_{ik}^{\tan}(\gamma) \end{bmatrix} := \begin{bmatrix} C_k(\gamma_0) \\ D_{ik}(\gamma_0) \end{bmatrix} + \begin{bmatrix} \frac{\partial C_k(\gamma)}{\partial\gamma}|_{\gamma=\gamma_0} \\ \frac{\partial D_{ik}(\gamma)}{\partial\gamma}|_{\gamma=\gamma_0} \end{bmatrix}(\gamma - \gamma_0)$$

$$\begin{bmatrix} C_k^{\text{sec}}(\gamma) \\ D_{ik}^{\text{sec}}(\gamma) \end{bmatrix} := \begin{bmatrix} C_k(S^0), ...C_k(S^{4d}) \\ D_{ik}(S^0), ...D_{ik}(S^{4d}) \end{bmatrix}\begin{bmatrix} S^0, ..., S^{4d} \\ 1, ..., 1 \end{bmatrix}^{-1}\begin{bmatrix} \gamma \\ 1 \end{bmatrix}.$$

$$(10)$$

Here $k = 1, 2, \forall\gamma_0 \in S$. As shown above, (9) is a linear programming problem that can be solved using various methods, including the simplex algorithm [Klee and Minty, 1972] and the interior algorithm [Kojima et al., 1989, Nesterov and Nemirovskii, 1994].

**4) Global_error()**: This function is to terminate PI-SFP via estimating the order of the error with respect to $n$. Recall that in **Step 5** of Algorithm. 1, we always select the $\tilde{S}_k$ with the lowest $\underline{f_S(Y_x = y)}$ in the $k$−th iteration. This strategy guarantees (see Appendix A.5 for more details)

$$\underline{f_{\tilde{S}_k}(Y_x = y)} \leq \min_{S\in\mathcal{S}_k} \underline{f_S(Y_x = y)} = \underline{f(Y_x = y)}, \quad (11)$$

i.e., all the $\underline{f_{\tilde{S}_k}(Y_x = y)}$'s are lower bounds of $\underline{f(Y_x = y)}$, and thus $\underline{f_{opt}^n(Y_x = y)} \leq \underline{f(Y_x = y)}$. From this, we further have that, in the $n$-th iteration, for any $k \in \{0, \cdots, n\}$,

$$0 \leq \underline{f(Y_x = y)} - \underline{f_{opt}^n(Y_x = y)}$$
$$\leq \min_{S\in\mathcal{S}_k} \underline{f_S(Y_x = y)} - \underline{f_{\tilde{S}_k}(Y_x = y)} \quad (12)$$
$$\leq \underline{f_{\tilde{S}_k}(Y_x = y)} - \underline{f_{\tilde{S}_k}(Y_x = y)}.$$

Also see Appendix A.5 for details. This allows us to calculate an error bound via targeting

$$\min_{0\leq k\leq n}\left\{\overline{f_{\tilde{S}_k}(Y_x = y)} - \underline{f_{\tilde{S}_k}(Y_x = y)}\right\}. \quad (13)$$

Since the bound of $\overline{f_{\tilde{S}_k}(Y_x = y)} - \underline{f_{\tilde{S}_k}(Y_x = y)}$ is dominated by the diameter of the simplex $\tilde{S}_k$, i.e., $dia(\tilde{S}_k)$, we aim to get an order of (13) based on the order of the smallest $dia(\tilde{S}_k)$ with respect to $n$. As shown in Eqn (A.48) in Appendix A.5, this order is controlled by the length $L_n$ of the longest nested subsequence of $\{\tilde{S}_k\}_{k=0}^n$, which is summarized as Algorithm 3.

---

[7]If $IR_\Gamma \cap S = \emptyset$, then $\underline{f_S(Y_x = y)} = +\infty$.

**Algorithm 3:** Procedure to estimate the current convergence (**Global_error**).

**Input:** Collections of simplex partitions in each iteration till $n-$th iteration: $\tilde{S}_0, \cdots, \tilde{S}_n$.

**Output:** An estimate of the global error.

1 Let $\{\tilde{S}_{i_k}\}_{k=1}^{L_n}$ be the (longest) subsequence of $\{\tilde{S}_k\}_{k=0}^n$ such that each $\tilde{S}_{i_{j+1}}$ is partitioned from $\tilde{S}_{i_j}$ for $j = 0, 1, \cdots, L_n - 1$, where $L_n$ is the length of this subsequence;

2 Return the global error estimate $(\frac{\sqrt{3}}{2})^{\lfloor \frac{L_n}{4d} \rfloor}$.

## 5 THEORETICAL ANALYSIS

This section delves into the theoretical properties of PI-SFP. First, we examine the general convergence rate of PI-SFP concerning $L_n$ (see Theorem. 1). Then, we demonstrate that PI-SFP can be extended from computing $f(Y_x = y)$ to the general ACE case.

**Assumption 2** *(Positivity) $\mathscr{P}$ is a set of $P(\boldsymbol{W} \mid \boldsymbol{U})$ guaranteeing each compatible solution $P(\boldsymbol{U}, \boldsymbol{X} = x)$ to be positive definite. Namely, $\exists \delta > 0$, such that $\forall \phi = (\boldsymbol{\theta}, \boldsymbol{\psi}, \boldsymbol{\omega}) \in IR_\phi$, we have $\boldsymbol{\psi} \geq \delta * \mathbf{1}_{1*d} > \mathbf{0}_{1*d}$.*

It is a fairly broad and reasonable assumption in practice, just in order to ensure that the denominator in the original formulation is not too small to facilitate the calculation. Under this assumption, we have $\psi_i^o < \frac{1}{\delta}$ in (3) and $\psi^l > \delta$ in (8). Hence we have $sup_{\gamma \in IR_\Gamma} \|\gamma\|_{+\infty} < +\infty$ and $dia(S_0) < +\infty$ respectively. In addition, when Assumption 2 is violated, we propose an alternative PI-SFP refer readers to Appendix A.3 for more information. On this basis, we formally collate the previous analysis as follows:

**Theorem 1** *Under Assumption. 1–2, PI-SFP concentrates around the target value $f(Y_x = y)$ at the $O((\frac{3}{4})^{\lfloor \frac{L_n}{4d} \rfloor})$ rate. Specifically, $\mid f_{opt}^n(Y_x = y) - f(Y_x = y) \mid \leq A(\frac{3}{4})^{\lfloor \frac{L_n}{4d} \rfloor} dia(S_0)^2$, where $A = A_1 + A_2 + A_3 < +\infty$, $A_1 = \max_{\gamma \in S_0} \frac{2(\sqrt{2}+1)\sqrt{d}}{\delta} \|\frac{\partial(C_1(\gamma) - C_2(\gamma))}{\partial \gamma}\|$, $A_2 = \max_{\gamma \in S_0} \|\frac{\partial^2 C_1(\gamma)}{\partial \gamma^2}\|_F$, $A_3 = \frac{1}{2} \max_{\gamma \in S_0} \|\frac{\partial^2 C_2(\gamma)}{\partial \gamma^2}\|_F$. Here $\| \cdot \|$ denotes the Euclidean norm, and $\| \cdot \|_F$ denotes the Frobenius norm. $L_n \in [\lfloor log(n) \rfloor + 1, n]$ is the length of the longest nested sequence till $n-$th iteration. Moreover, $\lim_{n \to +\infty} f_{opt}^n(Y_x = y) = f(Y_x = y)$.*

Theorem. 1 states that PI-SFP converges to $f(Y_x = y)$ with the growing length of the longest nested sequence, and will approach it in the infinite case. We relegate the proof to Appendix A.5 and reserve a brief summary. First, $f(Y_x = y)$ is equal to $f_{S_0}(Y_x = y)$ via constructing an original enclosure $S_0$ in (8). Second, $f_{S_0}(Y_x = y)$ is substituted

with $\min_{S \in \mathcal{S}_k} f_S(Y_x = y)$ in the $k$-th iteration by bisection. Third, each $f_S(Y_x = y)$ is lower bounded by (9), namely we have $\forall S \in \mathcal{S}_k, f_S(Y_x = y) \geq \underline{f_S}(Y_x = y)$. Finally, $\tilde{S}_k$ with the lowest bound $\min_{S \in \mathcal{S}_k} \underline{f_S}(Y_x = y)$ is gathered as $\{\tilde{S}_k\}_{k=0}^n$ in order to formulate $f_{opt}^n(Y_x = y)$ (see **Step 10** in Algorithm 1). The asymptotic error can be bounded by (12)-(13). In conclusion, these four steps correspond to the four functions in the above section in order.

Noteworthy, it is well beyond the scope of this paper to theoretically estimate $L_n$ w.r.t $n$, both empirically and theoretically. We refer readers to Appendix A.7.5 for detailed comment. In this comment, we also figure out a conjecture upon finiteness of regular simplicial partitions, which is our extra contribution. Moreover, our method and theorem could be naturally extended to the ACE case, which is detailed in Appendix A.6 for space limitation.

## 6 SIMULATIONS AND REAL-WORLD EXPERIMENTS

In this section, we perform experiments to demonstrate the efficacy of PI-SFP, aiming to address two key questions: 1) Can PI-SFP effectively manage the partially observable $P(\boldsymbol{W} \mid \boldsymbol{U})$, a scenario not previously explored in the literature, and generate informative bounds? 2) How does PI-SFP's convergence rate manifest in practical applications? Due to space constraints, some visualizations are deferred to Appendix A.8.

### 6.1 SIMULATIONS

**Experiment settings** We refer to the case presented in Section. 3, as shown in Equation. (A.5), and generalize our findings, with a specific focus on Fig. 1(a). We address an intriguing and universal situation referred to as 'information leakage,' where the information of $\boldsymbol{U}$ is regularly retained by $\boldsymbol{W}$ but incurs loss during transmission. Formally, we claim $P(W = w_i \mid U = u_i) \geq 1 - \varepsilon, \varepsilon \in (0, 1)$. To make the experiment simple and representative, we consider the binary cases of $\boldsymbol{W}, \boldsymbol{U}, \boldsymbol{X}$. On this basis, the construction is $P(\boldsymbol{W} \mid \boldsymbol{U}) := (1 - \varepsilon)\boldsymbol{I}_{2*2} + \varepsilon \boldsymbol{J}_{2*2}$ and $P(\boldsymbol{W} \mid \boldsymbol{U}) := (1 - \varepsilon)\boldsymbol{I}_{2*2}, \varepsilon \in (0, 1)$. The construction of $f(\boldsymbol{Y}, \boldsymbol{W}, \boldsymbol{X})$ still follows (A.5)[8]. Moreover, we set the iteration number as 1000.

**Experiment result** The simulation results, shown in Table. 1 and Fig.3, indicate that PI-SFP successfully finds optimal solutions and values, with a fast convergence rate within 1000 iterations. At the beginning step, estimation errors increase as $\varepsilon$ increases, but they remain under control

---

[8]In order to avoid the ill-conditioned case for PYTHON 3.8.5, we make a rather broad restriction that elements of $P(\boldsymbol{U}, x)$ are at least $1e^{-2}$ in all cases (Assumption. 2)

| $\varepsilon$ | $\Phi$ | | | | | | PI-SFP result | $f(Y_x = y)$ (Ground Truth) | Error |
|---|---|---|---|---|---|---|---|---|---|
| | $\theta_1$ | $\theta_2$ | $\psi_1$ | $\psi_2$ | $\omega_1$ | $\omega_2$ | | | |
| 0.1 | 0.067 | 0.133 | 0.261 | 0.239 | 0.333 | 0.167 | 0.370 | 0.372 | 0.548% |
| 0.2 | 0.050 | 0.150 | 0.262 | 0.238 | 0.375 | 0.125 | 0.350 | 0.351 | 0.285% |
| 0.3 | 0.029 | 0.171 | 0.264 | 0.236 | 0.429 | 0.072 | 0.298 | 0.301 | 0.997% |
| $\geq 0.4$ | 0.001 | 0.199 | 0.310 | 0.190 | 0.500 | 0.000 | 0.200 | 0.205 | 2.439% |

TABLE 1: *Simulation results (the upper bound is symmetric). The middle column $\Phi$ denotes the optimal solution within iteration $10^6$. The ground truth is approached via $10^6$ Monte-Carlo sampling. Our PI-SFP result decreases monotonically with the increasing $\varepsilon$, since the feasible region of latent variables $\Phi$ is gradually enlarged with $\varepsilon$. Detailed visulization of convergence rate is shown in Fig. 3 (Appendix A.8).*

by the theoretical error guaranteed by Theorem.1. During iterations, estimation errors converge quickly to the real $f(Y_x = y)$. The ground truth decreases as the feasible region of $\Phi$ increases, i.e., when $\varepsilon$ increases. Moreover, when $\varepsilon \geq 0.4$, we observe that $f(Y_x = y)$ achieves its minimum value Pearl [2009b] of $f(y, x) = 0.200$.

## 6.2 REAL-WORLD APPLICATIONS

In the simulation experiments, we have demonstrated that PI-SFP can quickly converge to the valid bound; furthermore, in the real experiments in this section, we show that the valid bound generated by PI-SFP can more effectively substantiate the causal relationships in the real world compared with previous methods. Specifically, we re-analyze the Zika Virus outbreak dataset [Taddeo et al., 2022, Tchetgen et al., 2024] in the most-related literature [Tchetgen et al., 2023]. Our PI-SFP result exhibits a more significant adverse effect from Zika Virus to the birth rate, which is more aligned with the well-known scientific hypothesis [Castro et al., 2018] compared with the previous literature [Tchetgen et al., 2023]. We defer the experimental details to Appendix A.8 due to space limitation.

## 7 JUSTIFICATION OF ASSUMPTIONS AND FURTHER DISCUSSIONS

In this section, for the core partial observability assumption (Assumption.1), we analyze its necessity, generalizability, and verifiability. First, the necessity is supported by the following lemma. It leads to an interesting and counter-intuitive *negative result*: informative proxies (namely transition matrix $P(\boldsymbol{W} \mid \boldsymbol{U})$ is reversible) *do not guarantee* informative bounds (instead of the vanilla bound);

**lemma 1** *Assume that $[P(\boldsymbol{W} \mid \boldsymbol{U}), \overline{P(\boldsymbol{W} \mid \boldsymbol{U})}] = [\boldsymbol{0}_{d_w * d_u}, \boldsymbol{1}_{d_w * d_u}]$, and $f(\boldsymbol{U}, x) > \boldsymbol{0}$. We consider the whole set of $f(y, \boldsymbol{W}, \boldsymbol{U}, \boldsymbol{X})$ which is within $\widetilde{\mathcal{F}}$ and is additionally compatible with two observed distributions $f(\boldsymbol{W}, \neg x) > \boldsymbol{0}_{d_w * 1}$, $f(y, \boldsymbol{W}, x) > \boldsymbol{0}_{d_w * 1}$ by an unknown $P(\boldsymbol{W} \mid \boldsymbol{U})$. Then (i) The tight lower bound of $f(Y_x = y)$*
is vanilla, namely $f(y, x)$. (ii) If $P(\boldsymbol{W} \mid \boldsymbol{U})$ is restricted to be left-reversible and $f(\boldsymbol{W} \mid \neg x) \neq f(\boldsymbol{W} \mid x, y)$, then the tight lower bound of $f(Y_x = y)$ is still the vanilla $f(y, x)$. (iii) If $P(\boldsymbol{W} \mid \boldsymbol{U})$ is restricted to be left-reversible and $f(\boldsymbol{W} \mid \neg x) = f(\boldsymbol{W} \mid x, y)$, then $f(Y_x = y)$ is lower bounded by another vanilla bound $f(y \mid x)$.

The proof is in Appendix A.7.1. This lemma extends the well-known inequality $f(Y_x = y) \geq f(y, x)$ [Pearl, 2009b] to single proxy control. Lemma 1 sufficiently indicates that partial observability Assumption 1 ([Kuroki and Pearl, 2014, Greenland, 2005]), instead of the reversibility assumption in the previous literature (e.g., [Miao and Tchetgen, 2018]), is more necessary for partial identification in most cases.

Furthermore, for genealizability, verifiability and practical correspondence of Assumption 1, we refer readers to arguments in Appendix A.7.1. Moreover, we provide algorithm comparison and algorithm acceleration in Appendix A.7.2, discuss graph structure extension in Appendix A.7.3, and then extend to the continuous case of confounders in Appendix A.7.4.

## 8 CONCLUSIONS

In this paper, we highlight the limitations and strict assumptions of the transfer matrix $P(\boldsymbol{W} \mid \boldsymbol{U})$ through practical examples, emphasizing that exact observability and reversibility are often not feasible in real-world scenarios. Based on this, we propose a novel PI-SFP framework that achieves a valid bound for the causal effect, even with only partial observability of $P(\boldsymbol{W} \mid \boldsymbol{U})$. To achieve it, we employ deformation techniques in DC programming and implement a branch-and-bound method. We offer a theoretical analysis of the mathematical reasons behind the lack of tight bounds and provide sufficient and necessary conditions to determine if the bounds are tight. We also conduct a convergence rate analysis of PI-SFP. Furthermore, we provide specific convergence rate analysis for these methods. We also provide a fundamental negative result that informative proxies might not yield informative partial identification bounds.

Our paper has initiated new research trajectories, specifically

focusing on the proximal partial identification with broader confounding proxy information. An additional avenue for exploration could involve evaluating the performance of PI-SFP under more intricate partial observability assumptions. Furthermore, it would be promising for in-depth investigation to extend our single-proxy control scheme to encompass double-proxy control and other causal graphs. These avenues remain further exploration in our future research.

## ACKNOWLEDGEMENT

The research was partially completed while Zhiheng Zhang was a student intern at Shanghai Qi Zhi Institute. He is grateful to Professor Yuhao Wang for suggestions.

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

| Literature | Tools | | | Assumptions | | |
|---|---|---|---|---|---|---|
| | Valid Instrument | Negative exposure | Negative outcome | Reversibility Completeness | Bridge function | Observability of $P(\boldsymbol{W} \mid \boldsymbol{U})$ |
| [Balke and Pearl, 1994] [Kitagawa, 2009] | ✓ | ✗ | ✗ | ✗ | ✗ | ✗ |
| [Kuroki and Pearl, 2014](1) [Rothman et al., 2008] [Lee and Bareinboim, 2020] | ✗ | ✗ | ✓ | ✓ | ✗ | ✓[a] |
| [Kuroki and Pearl, 2014](2) [Nagasawa, 2018] | ✗ | ✓ | ✓ | ✓ | ✗ | ✗ |
| [Miao et al., 2018] [Shi et al., 2020] [Singh, 2020] [Cui et al., 2020] [Tchetgen et al., 2020] [Deaner, 2018] | ✗ | ✓ | ✓ | ✓ | ✓ | ✗ |
| [Kallus et al., 2021] | ✗ | ✓ | ✓ | ✗ | ✓ | ✗ |
| **Our paper** | ✘ | ✘ | ✔ | ✘ | ✘ | ✔[b] |

TABLE 2: *Tools and assumptions of previous literature on partial identification. [Kuroki and Pearl, 2014](1) is without $\boldsymbol{Z}$, while (2) is with $\boldsymbol{Z}$.*

---

[a] $P(\boldsymbol{W} \mid \boldsymbol{U})$ is assumed to be reversible and explicitly, totally observed.
[b] In our paper, $P(\boldsymbol{W} \mid \boldsymbol{U})$ only needs be partially bounded.

In the appendices, we provide supplementary material and proofs for our main text. Appendices A.1-A.2 contain proofs for propositions. In Appendix A.1, we establish that $\mathcal{F} \subseteq \widetilde{\mathcal{F}}$. In Appendix A.2, we demonstrate that the bound is tight under certain conditions.

Appendix A.3 is dedicated to discussing the assumption. We explore cases where Assumption 2 does not hold.

Appendix A.5 contains the main results. Firstly, we show that the original simplex $S_0$ encloses our identification region. Secondly, we prove that the original optimization problem can be transformed into a set of sub-problems in the reduced space. Thirdly, we demonstrate our construction to transfer the original nonlinear optimization problem to the weaker linear case. Finally, we prove that our algorithm converges to the global optimal solution at an exponential rate.

Appendix A.6 extends our result from $f(Y_x = y)$ to the more general ACE.

Appendix A.7 is dedicated to extensions. We discuss 1) the previous assumptions in the original literature, 2) auxiliary acceleration strategies, 3) extension of graph structure, and 4) extension to the continuous confounding.

## A.1 THE PROOF OF PROPOSITION 1

According to Assumption. 1, by integration, we can also directly claim that if $f(y, \boldsymbol{U}, \boldsymbol{W}, \boldsymbol{X}) \in \mathcal{F}$, then

$$
\begin{aligned}
\underline{P(\boldsymbol{W} \mid \boldsymbol{U})}\boldsymbol{\theta} \leq P(y, \boldsymbol{W}, x) \leq \overline{P(\boldsymbol{W} \mid \boldsymbol{U})}\boldsymbol{\theta}, \forall x \in X. \\
\underline{P(\boldsymbol{W} \mid \boldsymbol{U})}\boldsymbol{\psi} \leq P(\boldsymbol{W}, x) \leq \overline{P(\boldsymbol{W} \mid \boldsymbol{U})}\boldsymbol{\psi}, \forall x \in X. \\
\underline{P(\boldsymbol{W} \mid \boldsymbol{U})}\boldsymbol{\omega} \leq f(\boldsymbol{W}, \neg x) \leq \overline{P(\boldsymbol{W} \mid \boldsymbol{U})}\boldsymbol{\omega}, \forall x \in X.
\end{aligned} \tag{A.1}
$$

Thus

$$
\begin{bmatrix} -\boldsymbol{I_{d*d}} \\ \boldsymbol{I_{d*d}} \end{bmatrix} \begin{bmatrix} f(y, \boldsymbol{W}, x)^T \\ f(\boldsymbol{W}, x)^T \\ f(\boldsymbol{W}, \neg x)^T \end{bmatrix}^T - \begin{bmatrix} -\overline{P(\boldsymbol{W} \mid \boldsymbol{U})} \\ \underline{P(\boldsymbol{W} \mid \boldsymbol{U})} \end{bmatrix} \phi \geq \boldsymbol{0}. \tag{A.2}
$$

Combined with the natural that $\theta_i \in [0, P(y,x)], \psi_i \in [0, P(x)], \omega_i \in [0, P(\neg x)], i = 1, 2, ...d$, we have $f(y, \boldsymbol{U}, \boldsymbol{W}, \boldsymbol{X}) \in \widetilde{\mathcal{F}}$. In conclusion, we claim $\mathcal{F} \subseteq \widetilde{\mathcal{F}}$.

## A.2 THE PROOF OF PROPOSITION 2

As the optimal solution $\phi_{opt}$ satisfies the constraint (2) in Proposition. (2), we can equivalently claim that $\phi_{opt}$ is compatible with some $f(y, \boldsymbol{W}, \boldsymbol{U}, \boldsymbol{X})$ which satisfies $f(y, \boldsymbol{W}, \boldsymbol{U}, \boldsymbol{X}) \in \mathcal{F}$. On this basis, the original formulation can be transformed with stricter constraints but equal minimum optimal value, namely that from

$$\min f(y, x) + \sum_{i=1}^{d} \frac{1}{\psi_i} \theta_i \omega_i, \text{ subject to: } f(y, \boldsymbol{U}, \boldsymbol{W}, \boldsymbol{X}) \in \widetilde{\mathcal{F}} \tag{A.3}$$

to

$$\min f(y, x) + \sum_{i=1}^{d} \frac{1}{\psi_i} \theta_i \omega_i, \text{ subject to: } f(y, \boldsymbol{U}, \boldsymbol{W}, \boldsymbol{X}) \in \widetilde{\mathcal{F}} \cap \mathcal{F} = \mathcal{F}. \tag{A.4}$$

This is equal to the original (2). Hence $\underline{f(Y_x = y)}$ is the tight lower bound of $f(Y_x = y)$ under constraint. 2.

By contrast, if the constraint (2) does not hold, then any $f(y, \boldsymbol{U}, \boldsymbol{W}, \boldsymbol{X})$ compatible with $\phi_{opt}$ will be within $\mathcal{F}^c \cap \widetilde{\mathcal{F}}$. In another word, the minimum value of the original formulation will be lower than that of Eqn (2), and the bound $\underline{f(Y_x = y)}$ is not tight. Proved.

We further provide an instance that our bound is tight:

$$\begin{bmatrix} f(Y = y, \boldsymbol{W}, x)^T \\ f(Y = y, \boldsymbol{W}, \neg x)^T \\ f(Y \neq y, \boldsymbol{W}, x)^T \\ f(Y \neq y, \boldsymbol{W}, \neg x)^T \end{bmatrix} = \begin{bmatrix} 0.08 & 0.12 \\ 0.15 & 0.1 \\ 0.18 & 0.12 \\ 0.15 & 0.1 \end{bmatrix}, \begin{bmatrix} \overline{P(\boldsymbol{W} \mid \boldsymbol{U})} \\ \underline{P(\boldsymbol{W} \mid \boldsymbol{U})} \end{bmatrix} = \begin{bmatrix} 0.6\boldsymbol{I_{2*2}} + 0.4\boldsymbol{J_{2*2}} \\ 0.6\boldsymbol{I_{2*2}} \end{bmatrix}. \tag{A.5}$$

Here $\boldsymbol{W}, \boldsymbol{U}, \boldsymbol{X}$ are all binary, and $\boldsymbol{I_{n*n}}, \boldsymbol{J_{n*n}}$ denote the $n$ dimensional identity matrix and all-ones matrix respectively. we can verify one of the optimal solutions $\phi_{opt} = [0 \; 0.2 \; 0.3 \; 0.2 \; 0.5 \; 0]^T$. The corresponding $f(y, \boldsymbol{U}, \boldsymbol{W}, \boldsymbol{X})$ satisfying Eqn (2) exists, whose explicit form is detailed as follows:

$$\begin{bmatrix} f(Y = y, W = w_1, \boldsymbol{U}, \boldsymbol{X}) & f(Y \neq y, W = w_1, \boldsymbol{U}, \boldsymbol{X}) \\ f(Y = y, W = w_2, \boldsymbol{U}, \boldsymbol{X}) & f(Y \neq y, W = w_2, \boldsymbol{U}, \boldsymbol{X}) \end{bmatrix} = \begin{bmatrix} 0 & 0.15 & 0.18 & 0.15 \\ 0.08 & 0 & 0 & 0 \\ 0 & 0.1 & 0.12 & 0.1 \\ 0.12 & 0 & 0 & 0 \end{bmatrix} \tag{A.6}$$

## A.3 FURTHER DISCUSSION ON ASSUMPTION. 2

In this section, we consider the case when Assumption. 2 does not hold. We propose a new version of PI-SFP. Recall that our objective function is:

$$\underline{f(Y_x = y)} = \min f(y, x) + \sum_{i=1}^{d} \frac{1}{\psi_i} \theta_i \omega_i, \text{ subject to: } f(y, \boldsymbol{U}, \boldsymbol{W}, \boldsymbol{X}) \in \widetilde{\mathcal{F}}, i.e., \phi \in IR_{\boldsymbol{\Phi}}. \tag{A.7}$$

In our main text, we let $\psi_i^o = \frac{1}{\psi_i}$. However, when we can not guarantee that $\exists \delta, \forall i \in \{1, 2, ...d\}, \psi_i \geq \delta$ (without Assumption. 2,), then $\psi_i^o$ may turn to infinity. On this basis, we introduce another algebraic distortion $\psi_i^o = \frac{\theta_i \omega_i}{\psi_i}$. Then the above programming can be transformed to:

$$\underline{f(Y_x = y)} = \min f(y, x) + \sum_{i=1}^{d} \psi_i^o \tag{A.8}$$

$$\text{subject to: } \boldsymbol{\gamma} \in IR_{\Gamma}, \text{ where } IR_{\Gamma} = \{\boldsymbol{\gamma} : \phi \in IR_{\boldsymbol{\Phi}}, \psi_i^o \psi_i = \theta_i \omega_i, \psi_i^o \leq C, i = 1, ...d.\}.$$

where $C$ is a local optimal value (a priori computed) of $\underline{f(Y_x = y)}$. On this basis, we can adopt the analogous strategy as in the traditional PI-SFP. Here the original $S_0$ is easy to be constructed since $\|\boldsymbol{\gamma}\|_{+\infty} < +\infty$.

Programming (A.8) can also be adopted under Assumption. 2. Compared with the traditional PI-SFP, firstly, programming (A.8) needs an a priori computed $C$ to upper bound $\psi_i^o$. Secondly, we will do linearization on $\psi_i^o \psi_i = \theta_i \omega_i$ instead of $\psi_i^o \psi_i = 1$, which is more complex. There is no guarantee of which version is better and we will explore it in the future work.

## A.4 VALID BOUND OF AVERAGE CAUSAL EFFECT (ACE)

The identification region of $f(Y_x)$ is constructed as follows.

$$IR_{F(Y_x)} = \{f(Y_x) : \int_{Y^L}^{Y^U} f(Y_x = y)dy = 1,$$

$$\forall y \in [Y^L, Y^U], f(y, \boldsymbol{U}, \boldsymbol{W}, \boldsymbol{X}) \in \mathcal{F}\}. \tag{A.9}$$

Then the valid bound of $ACE_{\boldsymbol{X} \to \boldsymbol{Y}}$ can be denoted as $[\underline{ACE_{\boldsymbol{X} \to \boldsymbol{Y}}}, \overline{ACE_{\boldsymbol{X} \to \boldsymbol{Y}}}]$:

$$\underline{ACE_{\boldsymbol{X} \to \boldsymbol{Y}}} \leq \min\{ACE_{\boldsymbol{X} \to \boldsymbol{Y}}$$

$$= \int_{X^L}^{X^U} \int_{Y^L}^{Y^U} f(Y_x = y)\pi(x)dxdy : f(Y_x) \in IR_{F(Y_x)}\},$$

$$\overline{ACE_{\boldsymbol{X} \to \boldsymbol{Y}}} \geq \max\{ACE_{\boldsymbol{X} \to \boldsymbol{Y}} \tag{A.10}$$

$$= \int_{X^L}^{X^U} \int_{Y^L}^{Y^U} f(Y_x = y)\pi(x)dxdy : f(Y_x) \in IR_{F(Y_x)}\}.$$

$[\underline{ACE_{\boldsymbol{X} \to \boldsymbol{Y}}}, \overline{ACE_{\boldsymbol{X} \to \boldsymbol{Y}}}]$ is the valid bound of ACE. In our paper, we aim to design an algorithm to seek the valid bound of $f(Y_x = y)$, and then extend our strategy from bounding $f(Y_x = y)$ to bounding ACE. Homoplastically, we only need to consider the optimization technique on the minimum case, and the maximum case will be symmetric.

## A.5 THE PROOF OF THEOREM. 1

**The sketch of proof** This is the main result of our paper. The main procedure are as follows:

$$| \underline{f(Y_x = y)} - \underline{f_{opt}^n(Y_x = y)} |$$

$$= | \underline{f(Y_x = y)} - \max_{k \in \{0,1,...n\}} \underline{f_{\tilde{S}_k}(Y_x = y)} | \qquad \text{Definition of } \underline{f_{opt}^n(Y_x = y)}$$

$$= \min_{k \in \{0,1,...n\}} | \underline{f(Y_x = y)} - \underline{f_{\tilde{S}_k}(Y_x = y)} |$$

$$\overset{(1)}{=} \min_{k \in \{0,1,...n\}} | \underline{f_{S_0}(Y_x = y)} - \underline{f_{\tilde{S}_k}(Y_x = y)} | \qquad \textbf{Initialization}$$

$$\overset{(2)}{=} \min_{k \in \{0,1,...n\}} | \min_{S \in \mathcal{S}_k} \underline{f_S(Y_x = y)} - \underline{f_{\tilde{S}_k}(Y_x = y)} | \qquad \textbf{Bisection} \tag{A.11}$$

$$\overset{*}{\leq} \min_{k \in \{0,1,...n\}} | \underline{f_{\tilde{S}_k}(Y_x = y)} - \underline{f_{\tilde{S}_k}(Y_x = y)} |$$

$$\leq | \underline{f_{\tilde{S}_{i_{L_n}}}(Y_x = y)} - \underline{f_{\tilde{S}_{i_{L_n}}}(Y_x = y)} |$$

$$\overset{(3)}{=} O(dia(\tilde{S}_{i_{L_n}})) \qquad \textbf{Bounding}$$

$$\overset{(4)}{=} O((\frac{\sqrt{3}}{2})^{\lfloor \frac{L_n}{4d} \rfloor}). \qquad \textbf{Global\_error}$$

$*$ is directly by **(2)** and we have previously mentioned it in Eqn (12). In the following demonstration, we mainly focus on procedure **(1)(2)(3)(4)**, corresponding to the algorithm part **Initialization**, **Bisection**, **Bounding**, **Global\_error** in order.

**(1) Initialization()** We will claim that $IR_\Gamma \subseteq S_0$.

**lemma 2** *The original $S_0$ satisfies $IR_\Gamma \subseteq S_0$, and thus $\underline{f(Y_x = y)} = \underline{f_{S_0}(Y_x = y)}$.*

**The proof of lemma.** (2) The simplex construction is as follows. $S_0$ is spanned by $\{S_0^0, S_0^1, ...S_0^{4d}\}$, where

$$S_0^i = \begin{cases} \boldsymbol{\gamma^l}, \ i = 0 \\ \boldsymbol{\gamma^l} + (\alpha - \mathbf{1_{1*4d}\gamma^l}) * \vec{e_i}, \ i \in \{1, 2, ...4d\} \end{cases} \text{, where } \boldsymbol{\gamma^l} = (\gamma_1^l, \gamma_2^l, ...\gamma_{4d}^l)^T, \tag{A.12}$$

where $S_0^i$ is the supporting vertices set described in our main text. For each $\gamma \in IR_\Gamma$, we attempt to provide a direct construction as follows:

$$\forall \gamma \in IR_\Gamma, \text{ we have } \gamma \stackrel{*}{=} \sum_{i=0}^{4d} \beta_i S_0^i, \ \beta_i = \begin{cases} 1 - \sum_{i=1}^{4d} \beta_i, i = 0 \\ \frac{\gamma \vec{e_i} - \gamma_i^l}{\alpha - \mathbf{1}_{1*4d}\gamma^l}, i = 1, 2, ...4d \end{cases}, \beta_i \in [0, 1], \tag{A.13}$$

To prove (A.13), we only need to prove the correctness of the equality $*$ and the fact $\beta_i \in [0, 1], \forall i = 0, 1, ...4d$.

First, we demonstrate the correctness of this construction.

$$\sum_{i=0}^{4d} \beta_i S_0^i = \beta_0 S_0 + \sum_{i=1}^{4d} \beta_i S_0^i$$

$$= \beta_0 \gamma^l + \sum_{i=1}^{4d} \beta_i \left( \gamma^l + (\alpha - \mathbf{1}_{1*4d}\gamma^l)\vec{e_i} \right) \qquad \text{(definition of } S_0^i)$$

$$= (1 - \sum_{i=1}^{4d} \beta_i)\gamma^l + \sum_{i=1}^{4d} \beta_i \left( \gamma^l + (\alpha - \mathbf{1}_{1*4d}\gamma^l)\vec{e_i} \right) \quad \text{(definition of } \beta_i) \tag{A.14}$$

$$= \gamma^l + \sum_{i=1}^{4d} \frac{\gamma \vec{e_i} - \gamma_i^l}{\alpha - \mathbf{1}_{1*4d}\gamma^l} \left( (\alpha - \mathbf{1}_{1*4d}\gamma^l)\vec{e_i} \right) \qquad \text{(definition of } \beta_i)$$

$$= \gamma^l + \left( \sum_{i=1}^{4d} \gamma \vec{e_i} - \gamma_i^l \right) \vec{e_i} = \gamma.$$

Second, we claim $\forall i \in \{1, ...4d\}, \beta_i \in [0, 1]$. Since we already have $\beta_i > 0, i = 1, 2, ...4d$ according to the construction of $\{\alpha, \gamma^l\}$, we only need to prove the left: $\beta_0 > 0$. Notice that

$$\sum_{i=1}^{4d} \beta_i = \sum_{i=1}^{4d} \frac{\gamma \vec{e_i} - \gamma_i^l}{\alpha - \mathbf{1}_{1*4d}\gamma^l} = \frac{\mathbf{1}_{1*4d}\gamma - \mathbf{1}_{1*4d}\gamma^l}{\alpha - \mathbf{1}_{1*4d}\gamma^l}. \tag{A.15}$$

Due to $\beta_0 = 1 - \sum_{i=1}^{4d} \beta_i$, it is equal to prove

$$\mathbf{1}_{1*4d}\gamma \leq \alpha = 1 + f(y, x) + \frac{d^2(\psi^l + \psi^u)^2}{4f(x)\psi^l\psi^u}, \tag{A.16}$$

where $\psi^l, \psi^u$ are identified in the main text. It is equivalent to

$$\sum_{i=1}^{d} \psi_i^o + \sum_{i=1}^{d} \theta_i + \sum_{i=1}^{d} \psi_i + \sum_{i=1}^{d} \omega_i \leq 1 + f(y, x) + \frac{d^2(\psi^l + \psi^u)^2}{4f(x)\psi^l\psi^u}, \tag{A.17}$$

namely that

$$\sum_{i=1}^{d} \psi_i^o \leq \frac{d^2(\psi^l + \psi^u)^2}{4f(x)\psi^l\psi^u}. \tag{A.18}$$

We only need prove the inequality (A.18). It is due to the fact $(\psi_i - \psi^l)(\frac{1}{\psi_i} - \frac{1}{\psi^u}) \geq 0$, namely $1 + \frac{\psi^l}{\psi^u} \geq \frac{\psi^l}{\psi_i} + \frac{\psi_i}{\psi^u}$. By which we have

$$(1 + \frac{\psi^l}{\psi^u})d \geq \psi^l \sum_{i=1}^{d} \frac{1}{\psi_i} + \frac{1}{\psi^u} \sum_{i=1}^{d} \psi_i \geq 2\sqrt{\frac{\psi^l}{\psi^u}}\sqrt{\sum_{i=1}^{d} \frac{1}{\psi_i}}\sqrt{f(x)}. \tag{A.19}$$

It is equal to

$$\sum_{i=1}^{d} \psi_i^o = \sum_{i=1}^{d} \frac{1}{\psi_i} \leq \frac{(\psi^u + \psi^l)^2 d^2}{4f(x)\psi^u\psi^l}, \text{ and thus } \sum_{i=1}^{4d} \beta_i \in [0, 1]. \tag{A.20}$$

On this basis, $\beta_0 = 1 - \sum_{i=1}^{4d} \beta_i \in [0, 1]$. Combining with $\beta_i \geq 0, i \in \{0, 1, ...4d\}$ and Eqn (A.13), we claim that $\forall \gamma \in IR_\Gamma$, we have $\gamma \in S_0$. Due to the arbitrary of $\gamma$, we have $IR_\Gamma \subseteq S_0$, and thus $\underline{f(Y_x = y)} = \underline{f_{S_0}(Y_x = y)}$.

**(2) Bisection()** We introduce the following lemma:

**lemma 3** *The partitioning set $\mathcal{S}_k$ satisfies $\underline{f_{S_0}(Y_x = y)} = \min\limits_{S \in \mathcal{S}_k} \underline{f_S(Y_x = y)}$.*

**The proof of lemma.** (3) By definition of bisection process, $\tilde{S}_k$ is bisectioned into $\tilde{S}_{k1}, \tilde{S}_{k2}$. Then

$$\mathcal{S}_{k+1} := \left( \mathcal{S}_k \setminus \tilde{S}_k \right) \cup \{ \tilde{S}_{k1}, \tilde{S}_{k2} \} \tag{A.21}$$

Hence we have $\cup_{S \in \mathcal{S}_k} S = \cup_{S \in \mathcal{S}_{k+1}} S, \forall k = 0, 1, ...$ Thus $S_0 = \cup_{S \in \mathcal{S}_k} S$, and we have

$$\underline{f_{S_0}(Y_x = y)} = \underline{f_{(\cup_{S \in \mathcal{S}_k} S)}(Y_x = y)} = \min\limits_{S \in \mathcal{S}_k} \underline{f_S(Y_x = y)}. \tag{A.22}$$

Hence we have proved.

**(3) Bounding()** We first introduce lemma. (4) and lemma. (5) for preparation, then the procedure **(3)** is proved by lemma. (6).

**lemma 4** *The decomposition of* (3) *can be established as Eqn* (5).

**The proof of lemma.** (4) Specifically, we give the explicit decomposition as follows, and the sub-script $cyc$ means the cycle of symbol set $[\psi_i^o, \theta_i, \omega_i]$:

$$\psi_i^o \theta_i \omega_i$$

$$= \left[ \frac{1}{2} \sum_{cyc} (\psi_i^o)^2 \theta_i + \frac{1}{2} \sum_{cyc} (\psi_i^o) \theta_i^2 + \sum_{i=1}^{d} \psi_i^o \theta_i \omega_i \right] - \frac{1}{2} \sum_{cyc} (\psi_i^o)^2 \theta_i - \frac{1}{2} \sum_{cyc} (\psi_i^o) \theta_i^2$$

$$= \left[ \frac{1}{6} (\sum_{cyc} \psi_i^o)^3 - \frac{1}{6} (\sum_{cyc} (\psi_i^o)^3) \right] - \frac{1}{2} \sum_{cyc} (\psi_i^o)^2 \theta_i - \frac{1}{2} \sum_{cyc} (\psi_i^o) \theta_i^2$$

$$= \left[ \frac{1}{6} (\sum_{cyc} \psi_i^o)^3 - \frac{1}{6} (\sum_{cyc} (\psi_i^o)^3) \right] + \frac{1}{2} \sum_{cyc} (\psi_i^o)^4 - \frac{1}{4} \sum_{cyc} (\psi_i^o)^4 - \frac{1}{4} \sum_{cyc} (\theta_i)^2 - \frac{1}{2} \sum_{cyc} (\psi_i^o)^2 \theta_i \tag{A.23}$$

$$+ \frac{1}{2} \sum_{cyc} (\psi_i^o)^2 - \frac{1}{4} \sum_{cyc} (\psi_i^o)^2 - \frac{1}{4} \sum_{cyc} (\theta_i)^4 - \frac{1}{2} \sum_{cyc} \psi_i^o \theta_i^2$$

$$= \left[ \frac{1}{6} (\sum_{cyc} \psi_i^o)^3 - \frac{1}{6} (\sum_{cyc} (\psi_i^o)^3) \right] + \frac{1}{2} \sum_{cyc} (\psi_i^o)^4 - \frac{1}{4} \sum_{cyc} ((\psi_i^o)^2 + \theta_i)^2 + \frac{1}{2} \sum_{cyc} (\psi_i^o)^2 -$$

$$\frac{1}{4} \sum_{cyc} (\psi_i^o + \theta_i^2)^2.$$

On this basis, if we choose

$$C_1(\boldsymbol{\gamma}) = \sum_{i=1}^{d} \left[ \frac{1}{6} (\sum_{cyc} \psi_i^o)^3 + \frac{1}{2} \sum_{cyc} (\psi_i^o)^4 + \frac{1}{2} \sum_{cyc} (\psi_i^o)^2 \right],$$

$$C_2(\boldsymbol{\gamma}) = \sum_{i=1}^{d} \left[ \frac{1}{6} \sum_{cyc} (\psi_i^o)^3 + \frac{1}{4} \sum_{cyc} [(\psi_i^o)^2 + \theta_i]^2 + \frac{1}{4} \sum_{cyc} [\psi_i^o + \theta_i^2]^2 \right], \tag{A.24}$$

then we have

$$\sum_{i=1}^{d} \psi_i^o \theta_i \omega_i = C_1(\boldsymbol{\gamma}) - C_2(\boldsymbol{\gamma}). \tag{A.25}$$

Here the Hessian matrix $\frac{\partial^2 C_1(\boldsymbol{\gamma})}{\partial^2 (\boldsymbol{\gamma})}$ and $\frac{\partial^2 C_2(\boldsymbol{\gamma})}{\partial^2 (\boldsymbol{\gamma})}$ is positive semi-definite:

$$\frac{\partial^2 C_1(\boldsymbol{\gamma})}{\partial^2 (\boldsymbol{\gamma})} = \frac{\partial^2 C_2(\boldsymbol{\gamma})}{\partial^2 (\boldsymbol{\gamma})} = [\boldsymbol{\gamma} + 6\boldsymbol{\gamma} \circ \boldsymbol{\gamma} + \mathbf{1_{4d*1}}] \circ \begin{bmatrix} \mathbf{1_{1*2d}} & \mathbf{0_{1*d}} & \mathbf{1_{1*d}} \end{bmatrix}^T \geq \mathbf{0_{4d*1}}, \tag{A.26}$$

where ∘ denotes the Hadamard product. Moreover,

$$\frac{\partial^2 D_{i1}(\boldsymbol{\gamma})}{\partial^2(\boldsymbol{\gamma})} = \frac{\partial^2 D_{i2}(\boldsymbol{\gamma})}{\partial^2(\boldsymbol{\gamma})} = \begin{bmatrix} \mathbf{1_{1*d}} & \mathbf{0_{1*d}} & \mathbf{1_{1*d}} & \mathbf{0_{1*d}} \end{bmatrix}^T \geq \mathbf{0_{4d*1}}. \tag{A.27}$$

$D_{i1}(\boldsymbol{\gamma}), D_{i2}(\boldsymbol{\gamma})$ are also positive semi-definite.

On this basis, we further give the upper and lower bound of the convex function as follows:

**lemma 5** *If function $F(\boldsymbol{\gamma})$ is differential and convex restricted by any simplex $S$, then $F^{tan}(\boldsymbol{\gamma}) \leq F(\boldsymbol{\gamma}) \leq F^{sec}(\boldsymbol{\gamma})$, where $\boldsymbol{\gamma_0} \in S$. In our paper, function $F(\cdot)$ can be chosen as $C_1(\cdot), C_2(\cdot), D_{i1}(\cdot), D_{i2}(\cdot)$, and $F^{tan}(\cdot), F^{sec}(\cdot)$ hold the same construction as in Formulation 10.* [9]

**The proof of lemma.** (5) The left part is intuitive. It is the tangent line equation of $F(\boldsymbol{\gamma})$. We only consider the right part by the convex property of $F(\boldsymbol{\gamma})$, whose construction is motivated by Pei and Zhu [2013]. We use $\boldsymbol{\gamma}_i, i = 1, 2, ...4d$ to denote the value of $\boldsymbol{\gamma}$ on each dimension ($\lambda_i \in [0, 1], \sum_{i=0}^{4d} \lambda_i = 1$):

$$\begin{aligned}
F(\boldsymbol{\gamma}) = F(\sum_{i=0}^{4d} \lambda_i S^i) &\leq \sum_{i=0}^{4d} \lambda_i F(S^i) = \sum_{i=0}^{4d} \lambda_i [F(S^0), F(S^1), ..., F(S^{4d})] \begin{bmatrix} S^0, ..., S^{4d} \\ 1, ..., 1 \end{bmatrix}^{-1} \begin{bmatrix} S^i \\ 1 \end{bmatrix} \\
&= [F(S^0), F(S^2), ..., F(S^{4d})] \begin{bmatrix} S^0, ..., S^{4d} \\ 1, ..., 1 \end{bmatrix}^{-1} \begin{bmatrix} \boldsymbol{\gamma} \\ 1 \end{bmatrix} = F^{sec}(\boldsymbol{\gamma}).
\end{aligned} \tag{A.28}$$

Hence we have proved our lemma.

On this basis, we can claim (9) provides the lower bound of $f_S(Y_x = y)$, namely $\underline{f_S(Y_x = y)} \leq f_S(Y_x = y)$. After the above difference-in-convex linear construction, we introduce the following lemma to approximate $f_S(Y_x = y)$ by $\underline{f_S(Y_x = y)}$:

**lemma 6** $\forall S, | \underline{f_S(Y_x = y)} - f_S(Y_x = y) | \leq A * dia(S)^2$, where $A = \max_{\gamma \in S_0} \|\frac{\partial(C_1(\gamma) - C_2(\gamma))}{\partial \gamma}\| \frac{2(\sqrt{2}+1)\sqrt{d}}{\delta} + \max_{\gamma \in S_0} \|\frac{\partial^2 C_1(\gamma)}{\partial \gamma^2}\|_F + \frac{1}{2} \max_{\gamma \in S_0} \|\frac{\partial^2 C_2(\gamma)}{\partial \gamma^2}\|_F < +\infty.$

**The proof of lemma.** (6) Since $dia(S_0) < +\infty$, we have that each element of $\gamma \in S_0$ can be bounded, namely $\|\gamma\|_{+\infty} < +\infty$. Then $\|\frac{\partial C_1(\gamma)}{\partial \gamma}\|, \|\frac{\partial C_2(\gamma)}{\partial \gamma}\|, \|\frac{\partial^2 C_1(\gamma)}{\partial \gamma^2}\|_F, \|\frac{\partial^2 C_2(\gamma)}{\partial \gamma^2}\|_F$ are all finite. Here $\|\cdot\|$ denotes the Euclidean norm, and $\|\cdot\|_F$ denotes the Frobenius norm.

If the corresponding optimal solution of $\underline{f_S(Y_x = y)}$ and $f_S(Y_x = y)$ are denoted as $\underline{\gamma}$ and $\underline{\underline{\gamma}}$ ($\underline{\gamma}, \underline{\underline{\gamma}} \in S$). Then according to lemma. (5), $| \underline{f_S(Y_x = y)} - f_S(Y_x = y) |$ can be bounded as follows:

$$\begin{aligned}
0 \leq &\underline{f_S(Y_x = y)} - f_S(Y_x = y) \\
= &| C_1(\underline{\gamma}) - C_1^{tan}(\underline{\underline{\gamma}}) - C_2(\underline{\gamma}) + C_2^{sec}(\underline{\underline{\gamma}}) | \\
\leq &| C_1(\underline{\gamma}) - C_1^{tan}(\underline{\underline{\gamma}}) - C_2(\underline{\gamma}) + C_2^{sec}(\underline{\underline{\gamma}}) | + | C_1(\underline{\underline{\gamma}}) - C_1(\underline{\gamma}) - C_2(\underline{\underline{\gamma}}) + C_2(\underline{\gamma}) | \\
\overset{*}{\leq} &\underbrace{| C_1(\underline{\underline{\gamma}}) - C_1^{tan}(\underline{\underline{\gamma}}) |}_{(1)} + \underbrace{| C_2(\underline{\underline{\gamma}}) - C_2^{sec}(\underline{\underline{\gamma}}) |}_{(2)} + \underbrace{| (C_1(\underline{\gamma}) - C_2(\underline{\gamma})) - (C_1(\underline{\underline{\gamma}}) - C_2(\underline{\underline{\gamma}})) |}_{(3)}
\end{aligned} \tag{A.29}$$

**item (1):** We consider the last line. The tangent line equation satisfies the following bound by Taylor expansion:

$$| C_1(\gamma) - C_1^{tan}(\gamma) | = O(\max_{\gamma \in S_0} \|\frac{\partial^2 C_1(\gamma)}{\partial \gamma^2}\|_F (dia(S))^2) = O(dia(S)^2), \tag{A.30}$$

---

[9] The matrix of the starting simplex $\begin{bmatrix} S_0^0, ..., S_0^{4d} \\ 1, ..., 1 \end{bmatrix}$ is reversible by the construction in lemma. (2). Moreover, the reversibility of $\begin{bmatrix} S^0, ..., S^{4d} \\ 1, ..., 1 \end{bmatrix}, S \in \mathcal{S}_k, k = 0, 1, ...$ still holds during bisection, since each bisection can be seen as a linear transformation between different columns.

**item (2)**: On the other hand, note that $\gamma = \sum_{j=0}^{4d} \lambda_j S^j$, here $\sum_{j=0}^{4d} \lambda_j = 1, \lambda_j \geq 0$:

$$\mid C_2(\gamma) - C_2^{\text{sec}}(\gamma) \mid = - [C_2(S^0), C_2(S^1), ...C_2(S^{4d})] \begin{bmatrix} S^0, ..., S^{4d} \\ 1, ..., 1 \end{bmatrix}^{-1} \begin{bmatrix} \sum_{i=0}^{4d} \lambda_i S^i \\ 1 \end{bmatrix} + C_2(\sum_{j=0}^{4d} \lambda_j S^j)$$

(A.31)

$$= \sum_{j=0}^{4d} \lambda_j C_2(S^j) - C_2(\sum_{j=0}^{4d} \lambda_j S^j).$$

We now aim to bound Eqn (A.31), inspired by [Budimir et al., 2001]. For simplicity, we use $\nabla$ to denote the derivative of a vector. Notice that the convex function has the property:

$$C_2(\sum_{i=0}^{4d} \lambda_j S^j) - C_2(S^j) \geq \langle \nabla C_2(S^j), \sum_{j=0}^{4d} \lambda_j S^j - S^j \rangle$$

(A.32)

By summation, we have

$$(\text{A.31}) = \sum_{j=0}^{4d} \lambda_j C_2(S^j) - C_2(\sum_{i=0}^{4d} \lambda_j S^j)$$

$$\leq \sum_{j=0}^{4d} \lambda_j \langle \nabla C_2(S^j), -\sum_{j=0}^{4d} \lambda_j S^j + S^j \rangle$$

(A.33)

$$= \sum_{j=0}^{4d} \lambda_j \langle \nabla C_2(S^j), S^j \rangle - \langle \sum_{j=0}^{4d} \lambda_j S^j, \sum_{j=0}^{4d} \lambda_j \nabla C_2(S^j) \rangle$$

(A.33) equals to

$$\frac{1}{2} \sum_{i=0}^{4d} \sum_{j=0}^{4d} \lambda_i \lambda_j \left[ [\langle \nabla C_2(S^j), S^j \rangle + \langle \nabla C_2(S^i), S^i \rangle] - [\langle \nabla C_2(S^j), S^i \rangle + \langle \nabla C_2(S^i), S^j \rangle] \right]$$

$$= \frac{1}{2} \sum_{i=0}^{4d} \sum_{j=0}^{4d} \lambda_i \lambda_j \langle S^i - S^j, \nabla C_2(S^i) - \nabla C_2(S^j) \rangle$$

$$\leq \frac{1}{2} \sum_{i=0}^{4d} \sum_{j=0}^{4d} \lambda_i \lambda_j \| S^i - S^j \| \| \nabla C_2(S^i) - \nabla C_2(S^j) \|$$

(A.34)

$$\leq \frac{1}{2} \sum_{i=0}^{4d} \sum_{j=0}^{4d} \lambda_i \lambda_j (\max_{\gamma \in S} \| \frac{\partial^2 C_2(\gamma)}{\partial \gamma^2} \|_F) \| S^i - S^j \|^2$$

$$\leq \frac{1}{2} \sum_{i=0}^{4d} \sum_{j=0}^{4d} \lambda_i \lambda_j (\max_{\gamma \in S} \| \frac{\partial^2 C_2(\gamma)}{\partial \gamma^2} \|_F) dia(S)^2$$

$$\leq \frac{1}{2} (\max_{\gamma \in S_0} \| \frac{\partial^2 C_2(\gamma)}{\partial \gamma^2} \|_F) dia(S)^2.$$

We have

$$0 \leq \sum_{j=0}^{4d} \lambda_j C_2(S^j) - C_2(\sum_{j=0}^{4d} \lambda_j S^j) \leq \frac{1}{2} \max_{\gamma \in S} \| \frac{\partial^2 C_2(\gamma)}{\partial \gamma^2} \|_F (dia(S))^2 = O((dia(S))^2).$$

(A.35)

Thus

$$\text{Eqn (A.31)} = \sum_{j=0}^{4d} \lambda_j C_2(S^j) - C_2(\sum_{j=0}^{4d} \lambda_j S^j) = O((dia(S))^2).$$

(A.36)

**item (3)** We introduce an auxiliary optimization problem as follows:

$$\min f(y,x) + C_1(\boldsymbol{\gamma}) - C_2(\boldsymbol{\gamma})$$
$$\text{subject to}: \phi \in IR_\Phi, D_{i1}(\boldsymbol{\gamma}) - D_{i2}(\boldsymbol{\gamma}) = 1,$$
$$\boldsymbol{\gamma} \in \{\boldsymbol{\gamma}' : \exists \boldsymbol{\gamma}'' \in S, \|\boldsymbol{\gamma}' - \boldsymbol{\gamma}''\| \le \frac{(\sqrt{2}+1)\sqrt{d}}{\delta} dia(S)^2\} \cap S_0. \tag{A.37}$$

Compared with the optimization problem of $f_S(Y_x = y)$ (by (A.37) with an additional constraint $\boldsymbol{\gamma} \in S$), (A.37) provides a relaxed constraint on $\boldsymbol{\gamma}$. We denote the optimal solution of (A.37) as $\underset{\sim}{\boldsymbol{\gamma}}$, and the optimal value as $\underset{\sim}{f_S}(Y_x = y)$.

On the one hand, (A.37) slightly relaxes the constraint $\boldsymbol{\gamma} \in S_0$. Namely for each $\underset{\sim}{\boldsymbol{\gamma}}$, there exists a corresponding $\boldsymbol{\gamma}'' \in S$ with a distance less than $\frac{Q}{\delta} dia(S)^2$. Hence

$$[C_1(\underset{\sim}{\boldsymbol{\gamma}}) - C_2(\underset{\sim}{\boldsymbol{\gamma}})] - [C_1(\boldsymbol{\gamma}) - C_2(\boldsymbol{\gamma})]$$
$$\le [C_1(\boldsymbol{\gamma}'') - C_2(\boldsymbol{\gamma}'')] - [C_1(\boldsymbol{\gamma}) - C_2(\boldsymbol{\gamma})]$$
$$\le \max_{\boldsymbol{\gamma} \in S_0} \|\frac{\partial(C_1(\boldsymbol{\gamma}) - C_2(\boldsymbol{\gamma}))}{\partial \boldsymbol{\gamma}}\| \left(\frac{(\sqrt{2}+1)\sqrt{d}}{\delta} dia(S)^2\right). \tag{A.38}$$

On the other hand, we consider the optimal solution $\underset{=}{\boldsymbol{\gamma}}$ of $f_S(Y_x = y)$. We identify the elements $\underset{=}{\boldsymbol{\gamma}}^T = ((\underset{=}{\boldsymbol{\psi^o}})^T, \underset{=}{\boldsymbol{\theta}}^T, \underset{=}{\boldsymbol{\psi}}^T, \underset{=}{\boldsymbol{\omega}}^T)$. Then we introduce an auxiliary solution as follows:

$$\underset{\sim}{\psi_i^o} = \frac{1}{\underset{=}{\psi_i}}, \underset{\sim}{\boldsymbol{\psi^o}} = (\underset{\sim}{\psi_1^o}, ... \underset{\sim}{\psi_d^o}), \underset{\sim}{\boldsymbol{\gamma}}' = ((\underset{\sim}{\boldsymbol{\psi^o}})^T, \underset{=}{\boldsymbol{\theta}}^T, \underset{=}{\boldsymbol{\psi}}^T, \underset{=}{\boldsymbol{\omega}}^T). \tag{A.39}$$

We will show that $\underset{\sim}{\boldsymbol{\gamma}}'$ is within the feasible region of (A.37). By identification in (A.39), the first row of constraints in (A.37) can be directly satisfied. Moreover, by Assumption. 2, we have

$$\|\underset{\sim}{\boldsymbol{\gamma}}' - \underset{=}{\boldsymbol{\gamma}}\| = \left(\sum_{i=1}^d (\frac{1}{\underset{=}{\psi_i}} - \underset{=}{\psi_i^o})^2\right)^{\frac{1}{2}}$$
$$\le \frac{1}{\delta}(\sum_{i=1}^d (\underset{=}{\psi_i}\underset{=}{\psi_i^o} - 1)^2)^{\frac{1}{2}}$$
$$= \frac{1}{\delta}(\sum_{i=1}^d (D_{i1}(\underset{=}{\boldsymbol{\gamma}}) - D_{i2}(\underset{=}{\boldsymbol{\gamma}}) - 1)^2)^{\frac{1}{2}} \tag{A.40}$$
$$\le \frac{\sqrt{d}}{\delta} \max_{i=1,...d} \left(1 - \left(D_{i1}(\underset{=}{\boldsymbol{\gamma}}) - D_{i2}(\underset{=}{\boldsymbol{\gamma}})\right)\right)$$
$$\le \frac{\sqrt{d}}{\delta} \max_{i=1,...d} \left[\left(D_{i1}^{sec}(\underset{=}{\boldsymbol{\gamma}}) - D_{i2}^{tan}(\underset{=}{\boldsymbol{\gamma}})\right) - \left(D_{i1}(\underset{=}{\boldsymbol{\gamma}}) - D_{i2}(\underset{=}{\boldsymbol{\gamma}})\right)\right].$$

Symmetrically, we have

$$\|\underset{\sim}{\boldsymbol{\gamma}}' - \underset{=}{\boldsymbol{\gamma}}\| \le \frac{\sqrt{d}}{\delta} \max_{i=1,...d} \left[\left(D_{i1}(\underset{=}{\boldsymbol{\gamma}}) - D_{i2}(\underset{=}{\boldsymbol{\gamma}})\right) - \left(D_{i1}^{tan}(\underset{=}{\boldsymbol{\gamma}}) - D_{i2}^{sec}(\underset{=}{\boldsymbol{\gamma}})\right)\right]. \tag{A.41}$$

By the same strategy in **item(1)-(2)**, and noticing the fact that

$$\max_{\boldsymbol{\gamma} \in S} \|\frac{\partial^2 D_{i1}(\boldsymbol{\gamma})}{\partial \boldsymbol{\gamma}^2}\|_F = 2, \max_{\boldsymbol{\gamma} \in S} \|\frac{\partial^2 D_{i2}(\boldsymbol{\gamma})}{\partial \boldsymbol{\gamma}^2}\|_F = \sqrt{2}, \tag{A.42}$$

(A.40) and (A.41) can be combined as

$$\|\underset{\sim}{\gamma}' - \underset{=}{\gamma}\| \leq \frac{\sqrt{d}}{\delta} \min\{\frac{1}{2} * 2 + \sqrt{2}, 2 + \frac{1}{2}\sqrt{2}\} dia(S)^2 = \frac{\sqrt{d}}{\delta}(\sqrt{2} + 1) dia(S)^2. \tag{A.43}$$

Hence we claim this $\underset{\sim}{\gamma}'$ is within the feasible region of (A.37). Then

$$\left[C_1(\underset{\sim}{\gamma}) - C_2(\underset{\sim}{\gamma})\right] - \left[C_1(\underset{=}{\gamma}) - C_2(\underset{=}{\gamma})\right] \leq \left[C_1(\underset{\sim}{\gamma}') - C_2(\underset{\sim}{\gamma}')\right] - \left[C_1(\underset{=}{\gamma}) - C_2(\underset{=}{\gamma})\right]$$

$$\leq \max_{\gamma \in S} \|\frac{\partial(C_1(\gamma) - C_2(\gamma))}{\partial \gamma}\| \frac{\sqrt{d}}{\delta}(\sqrt{2} + 1) dia(S)^2. \tag{A.44}$$

Combining (A.38) and (A.44), we have

$$| (C_1(\underset{=}{\gamma}) - C_2(\underset{=}{\gamma})) - (C_1(\underset{\sim}{\gamma}) - C_2(\underset{\sim}{\gamma})) | \leq \max_{\gamma \in S_0} \|\frac{\partial(C_1(\gamma) - C_2(\gamma))}{\partial \gamma}\| \frac{2\sqrt{d}}{\delta}(\sqrt{2} + 1) dia(S)^2. \tag{A.45}$$

**Combination of item(1)-(3)** Combining with Eqn (A.30) and Eqn (A.36), and recalling the bound in (A.29), we have:

$$\underline{f_S(Y_x = y)} \leq \underline{f_S(Y_x = y)} \leq \underline{f_S(Y_x = y)} + A * dia(S)^2. \tag{A.46}$$

In brief, we have $| \underline{\underline{f_S(Y_x = y)}} - \underline{f_S(Y_x = y)} | = O(dia(S)^2)$. Thus we have proved our lemma.

**Remark 1** *We can do enhancement in **Bounding** as follows. It is through taking advantage of the information from the parent simplex $pa(S)$* [10] *and encapsulating the above bounding strategy into a recursive form during partitioning.*

$$\underline{\underline{f_S(Y_x = y)}} = \max\{\textbf{\textit{Bounding}}(pa(S)), \underline{\underline{f_S(Y_x = y)}}\} \tag{A.47}$$

**(4) Global_error** For the final preparation, we introduce the bisection theorem:

**Theorem 2** *(Kearfott [1978], Theorem 3.1) When $\tilde{S}_{i_k}$ is bisectioned from $S_0$ by $k$ times, we have $dia(\tilde{S}_{i_k}) \leq (\frac{\sqrt{3}}{2})^{\lfloor \frac{k}{4d} \rfloor} dia(S_0)$.*

On this basis, notice that lemma. (6) holds on each iteration, and $dia(S_0) < +\infty$, then we have

$$| \underline{f_{\tilde{S}_{i_{L_n}}}(Y_x = y)} - \underline{\underline{f_{\tilde{S}_{i_{L_n}}}(Y_x = y)}} | \leq A((\frac{3}{4})^{\lfloor \frac{L_n}{4d} \rfloor}) = O((\frac{3}{4})^{\frac{L_n}{4d}}), \tag{A.48}$$

where $A$ is identified in our main text.

Until here we have proved procedure **(1)-(4)**, thus the main part of Theorem. 1 has been proved.

Additionally, consider the infinite case. Due to $L_n \geq log(n)$ (the worst case is that simplices set is bisectioned like a complete binary tree), we have $L \to +\infty$ when $n \to +\infty$, thus $\lim_{n \to +\infty} | \underline{f(Y_x = y)} - \underline{\underline{f^n_{opt}(f(Y_x = y))}} | = 0$. Done.

## A.6   EXTENSION TO THE ACE CASE

Taking advantage of PI-SFP, we can further achieve the valid bound of $ACE_{\boldsymbol{X} \to \boldsymbol{Y}}$. The above PI-SFP algorithm is to seek $\underline{f(Y_x = y)}$ when $x$ is fixed. We do further extension to consider all values of $\boldsymbol{X}$ simultaneously. In this sense, we reorganize (2) to bound ACE as follows:

---

[10] $S_1 = pa(S_2)$ denotes $S_2$ is bisectioned from $S_1$.

$$\min \sum_x \pi(x) \int_{Y^L}^{Y^U} y f(y,x) dy$$

$$+ \sum_x \pi(x) \sum_{i=1}^d \frac{\left( \int_{Y^L}^{Y^U} y f(y, u_i, x) dy \right) f(u_i, \neg x)}{f(u_i, x)} \tag{A.49}$$

$$\text{subject to: } f(y, \boldsymbol{U}, \boldsymbol{W}, \boldsymbol{X}) \in \mathcal{F}.$$

Using the same strategy as in Section. 3-4, we can achieve the valid bound of $ACE_{\boldsymbol{X} \to \boldsymbol{Y}}$ in (A.10),

We first illustrate the construction of (A.49):

$$\int_{X^L}^{X^U} \int_{Y^L}^{Y^U} f(Y_x = y) \pi(x) dx dy$$

$$= \int_{X^L}^{X^U} \int_{Y^L}^{Y^U} \sum_{i=1}^d \left( \frac{f(y, u_i, x) f(u_i, \neg x)}{f(u_i, x)} + f(y, x) \right) \pi(x) dx dy \tag{A.50}$$

$$= \sum_x \pi(x) \int_{Y^L}^{Y^U} y f(y, x) dy + \sum_x \pi(x) \sum_{i=1}^d \frac{\left( \int_{Y^L}^{Y^U} y f(y, u_i, x) dy \right) f(u_i, \neg x)}{f(u_i, x)}.$$

Let $X = \{x_1, x_2, ...x_{d_x}\}$. In this section, we extend PI-SFP method from bounding $f(Y_x = y)$ to bounding $ACE$. For simplicity, we extend the denotations in our main text as follows:

$$\begin{aligned}
\theta_{i|x} &= \int_{Y^L}^{Y^U} y f(y, U = u_i, x) dy, & \boldsymbol{\theta_x} &= (\theta_{1|x}, \theta_{2|x}, ...\theta_{d|x})^T, \\
\psi_{i|x} &= f(U = u_i, x), & \boldsymbol{\psi_x} &= (\psi_{1|x}, \psi_{2|x}, ...\psi_{d|x})^T, \\
\omega_{i|x} &= f(U = u_i, \neg x), & \boldsymbol{\omega_x} &= (\omega_{1|x}, \omega_{2|x}, ...\omega_{d|x})^T, \\
\psi_{i|x} \psi_{i|x}^o &= 1, & \boldsymbol{\psi_x^o} &= (\psi_{1|x}^o, \psi_{2|x}^o, ...\psi_{d|x}^o)^T, \\
\boldsymbol{\phi_x} &= (\boldsymbol{\theta_x}, \boldsymbol{\psi_x}, \boldsymbol{\omega_x}), & \boldsymbol{\gamma_x} &= \left( (\boldsymbol{\psi_x^o})^T, \boldsymbol{\theta_x}^T, \boldsymbol{\psi_x}^T, \boldsymbol{\omega_x}^T \right)^T.
\end{aligned} \tag{A.51}$$

On this basis, the independent variables are transformed to $\boldsymbol{\gamma} = (\boldsymbol{\gamma_{x_1}}, \boldsymbol{\gamma_{x_2}}, ...\boldsymbol{\gamma_{x_{d_x}}})$. Following the same strategy as in Section. 3 and Section. 4, we can relax the programming (A.49) in our main text as follows. It is a natural extension of (3) in Section. 3, by which we seek the valid bound of ACE:

$$\underline{ACE_{\boldsymbol{X} \to \boldsymbol{Y}}} = \min \sum_x \int_{Y^L}^{Y^U} \pi(x) y f(y,x) dy + \sum_x \sum_{i=1}^d \psi_{i|x}^o \theta_{i|x} \omega_{i|x} \pi(x),$$

$$\text{subject to } \forall x \in X, \psi_{i|x}^o \psi_{i|x} = 1, \boldsymbol{\phi_x} \in IR_{\boldsymbol{\Phi_x}} = IR_{\boldsymbol{\Phi_x}}^1 \cap IR_{\boldsymbol{\Phi_x}}^2, \tag{A.52}$$

where the set $IR_{\boldsymbol{\Phi}}^1$ is constructed as

$$IR_{\boldsymbol{\Phi_x}}^1 = \left\{ \boldsymbol{\phi_x} : \begin{bmatrix} -\boldsymbol{I_{d*d}} \\ \boldsymbol{I_{d*d}} \end{bmatrix} \begin{bmatrix} (\int_{Y^L}^{Y^U} y f(y, \boldsymbol{W}, x) dy)^T \\ f(\boldsymbol{W}, x)^T \\ f(\boldsymbol{W}, \neg x)^T \end{bmatrix}^T - \begin{bmatrix} -\overline{P(\boldsymbol{W} \mid \boldsymbol{U})} \\ \underline{P(\boldsymbol{W} \mid \boldsymbol{U})} \end{bmatrix} \boldsymbol{\phi_x} \geq 0 \right\}. \tag{A.53}$$

$\boldsymbol{I_{d*d}}$ is the $d * d$ identity matrix. Moreover, the set $IR_{\boldsymbol{\Phi}}^2$ indicates the natural constraints by default:

$$IR_{\boldsymbol{\Phi_x}}^2 = \left\{ \boldsymbol{\phi_x} : \begin{bmatrix} \boldsymbol{1_{1*d}} \boldsymbol{\theta_x} \\ \boldsymbol{1_{1*d}} \boldsymbol{\phi_x} \\ \boldsymbol{1_{1*d}} \boldsymbol{\omega_x} \end{bmatrix} = \begin{bmatrix} \int_{Y^L}^{Y^U} y f(y, x) dy \\ f(x) \\ f(\neg x) \} \end{bmatrix}, \forall i, \begin{cases} \theta_{i|x} \in [0, f(y, x)] \\ \phi_{i|x} \in (0, f(x)] \\ \omega_{i|x} \in [0, f(\neg x)] \end{cases} \right\}. \tag{A.54}$$

$\mathbf{1_{1*d}}$ is the $1 * d$ all-ones vector. Then (7) in our main text is extended as

$$\min \sum_x \pi(x)\{[C_1^{\tan}(\boldsymbol{\gamma}_x) - C_2^{\sec}(\boldsymbol{\gamma}_x)]\mathbb{1}_{\pi(x)>0} + [C_1^{\sec}(\boldsymbol{\gamma}_x) - C_2^{\tan}(\boldsymbol{\gamma}_x)]\mathbb{1}_{\pi(x)<0}\}, \tag{A.55}$$
$$\text{subject to } D_i^l(\boldsymbol{\gamma}_x) \leq 1, D_i^u(\boldsymbol{\gamma}_x) \geq 1, \forall i = 1, 2, ...d, \boldsymbol{\phi_x} \in IR_{\boldsymbol{\Phi_x}}.$$

Here the function $C_k^{\tan}(\boldsymbol{\gamma}_x), C_k^{\sec}(\boldsymbol{\gamma}_x), k = 1, 2, D_i^l(\boldsymbol{\gamma}_x), D_i^u(\boldsymbol{\gamma}_x), i = 1, 2, ...d$ are all following (10) in our main text. After this construction, we adopt the same simplicial partition strategy as in our main text.

## A.7 THE PROOF OF FURTHER DISCUSSIONS AND EXTENSIONS

This part is the supplement of the discussion in the main paper.

### A.7.1 Discussion 1: the proof of lemma. (1) and the justification of our Assumption 1

For simplification, the denotations $Y = y$, $x$ are simplified as $y$ and $x$, the denotation $\neg x$ is simplified as $x^c$, and $d_w$ is simplified as $\mathscr{W}$. Samely, we use $\boldsymbol{E_{i*i}}$ to denote the $i * i$ identity matrix, $\boldsymbol{J_{i,j}}$ to denote the $i * j$ all-ones matrix, and $\boldsymbol{0_{i*j}}$ to denote the $i * j$ all-zero matrix.

- **Conclusion 1:** The tight lower bound of $f(Y_x = y)$ is $f(y, x)$.

We divide it into two parts. On the one hand, if $\mathscr{W} \geq d$, $P(\boldsymbol{W} \mid \boldsymbol{U})$ can be constructed as follows.

$$P(\boldsymbol{W} \mid \boldsymbol{U}) = \begin{bmatrix} \overbrace{\boldsymbol{P_{11}}}^{m*m} & \vdots & \overbrace{\boldsymbol{P_{12}}}^{(d-m)*(d-m)} \\ \overbrace{\boldsymbol{P_{21}}}^{(\mathscr{W}-m)*m} & \vdots & \overbrace{\boldsymbol{P_{22}}}^{(\mathscr{W}-d+m)*(d-m)} \end{bmatrix}, \tag{A.56}$$

where $\boldsymbol{P_{11}}, \boldsymbol{P_{12}}, \boldsymbol{P_{21}}, \boldsymbol{P_{22}}$ are matrices whose upper brackets indicate their rows and columns ($m \in [1, d-1]$). Specifically,

$$\boldsymbol{P_{11}} = \sum_{i=1}^m f(W = w_i \mid y, x)\boldsymbol{E_{m*m}}, \quad \boldsymbol{P_{12}} = \sum_{i=1}^{d-m} f(W = w_i \mid x^c)\boldsymbol{E_{(d-m)*(d-m)}},$$
$$\boldsymbol{P_{21}} = \begin{bmatrix} f(W = w_{m+1} \mid y, x) \\ ... \\ f(W = w_{\mathscr{W}} \mid y, x) \end{bmatrix} \boldsymbol{J_{1*m}}, \quad \boldsymbol{P_{22}} = \begin{bmatrix} f(W = w_{d-m+1} \mid x^c) \\ ... \\ f(W = w_{\mathscr{W}} \mid x^c) \end{bmatrix} \boldsymbol{J_{1*(d-m)}}. \tag{A.57}$$

There is a solution for $f(y, \boldsymbol{U}, x), f(\boldsymbol{U}, x^c)$ respectively as

$$\frac{1}{\sum_{i=1}^m f(W = w_i \mid y, x)} \begin{bmatrix} f(y, W = w_1, x) \\ ... \\ f(y, W = w_m, x) \\ \boldsymbol{0_{(d-m)*1}} \end{bmatrix}, \quad \frac{1}{\sum_{i=1}^{d-m} P(W = w_i \mid x^c)} \begin{bmatrix} \boldsymbol{0_{m*1}} \\ P(W = w_1, x^c) \\ ... \\ P(W = w_{d-m}, x^c) \end{bmatrix}. \tag{A.58}$$

Due to $f(y, \boldsymbol{U}, x) \circ f(\boldsymbol{U}, x^c) = 0$ and the condition $f(\boldsymbol{U}, x) > \boldsymbol{0}$, we have

$$f(Y_x = y) = f(y, x) + \sum_{i=1}^d \frac{f(y, u_i, x)}{P(u_i, x)}P(u_i, x^c) = f(y, x). \tag{A.59}$$

On the other hand, if $\mathscr{W} < d$, we make adjustments on (A.56) ($m_1 + m_2 \leq \mathscr{W}$):

$$\begin{bmatrix} \overbrace{\boldsymbol{P_{11}}}^{m_1*m_1} & \vdots & \overbrace{\boldsymbol{P_{12}}}^{m_2*m_2} & \vdots & \overbrace{\boldsymbol{P_3}}^{\mathscr{W}*(d-m_1-m_2)} \\ \overbrace{\boldsymbol{P_{21}}}^{(\mathscr{W}-m_1)*m_1} & \vdots & \overbrace{\boldsymbol{P_{22}}}^{(\mathscr{W}-m_2)*m_2} & \vdots & \end{bmatrix}, \tag{A.60}$$

Specifically,

$$\boldsymbol{P_{11}} = \sum_{i=1}^{m_1} P(W = w_i \mid y, x)\boldsymbol{E_{m_1 * m_1}}, \ \boldsymbol{P_{12}} = \sum_{i=1}^{m_2} P(W = w_i \mid x^c)\boldsymbol{E_{m_2 * m_2}}.$$

$$\boldsymbol{P_{21}} = \begin{bmatrix} f(W = w_{m_1+1} \mid y, x) \\ ... \\ f(W = w_{\mathscr{W}} \mid y, x) \end{bmatrix} \boldsymbol{J_{1,m_1}}, \ \boldsymbol{P_{22}} = \begin{bmatrix} f(W = w_{m_2+1} \mid x^c) \\ ... \\ f(W = w_{\mathscr{W}} \mid x^c) \end{bmatrix} \boldsymbol{J_{1*m_2}}. \tag{A.61}$$

$$\boldsymbol{P_3} = \frac{1}{\mathscr{W}} \boldsymbol{J_{\mathscr{W}*(d-m_1-m_2)}}.$$

Analogously, there is a solution for $f(y, \boldsymbol{U}, x), f(\boldsymbol{U}, x^c)$ as follows respectively:

$$\frac{1}{\sum\limits_{i=1}^{m_1} f(W = w_i \mid y, x)} \begin{bmatrix} f(y, W = w_1, x) \\ ... \\ f(y, W = w_{m_1}, x) \\ \boldsymbol{0_{(d-m_1)*1}} \end{bmatrix}, \ \frac{1}{\sum\limits_{i=1}^{m_2} P(W = w_i \mid x^c)} \begin{bmatrix} \boldsymbol{0_{m_1*1}} \\ P(W = w_1, x^c) \\ ... \\ P(W = w_{m_2}, x^c) \\ \boldsymbol{0_{(d-m_1-m_2)*1}} \end{bmatrix}. \tag{A.62}$$

In this case, we also have $f(Y_x = y) = f(y, x) + \sum_{i=1}^{d} \frac{f(y,u_i,x)}{f(u_i,x)} f(u_i, x^c) = f(y, x)$. In conclusion, if no assumptions are imposed, we have $\min f(Y_x = y) = f(y, x)$. Proved.

- **Conclusion 2:** If $P(\boldsymbol{W} \mid \boldsymbol{U})$ is restricted to be left-reversible and $f(\boldsymbol{W} \mid \neg x) \neq f(\boldsymbol{W} \mid x, y)$, then the tight lower bound of $f(Y_x = y)$ is $f(y, x)$.

Without loss of generalization, we can assume that $\exists i_0 \in \{d, d+1, ...\mathscr{W}\}$, such that $f(W = w_{i_0} \mid \neg x) \neq f(W = w_{i_0} \mid x, y)$, or else we just need to relabel $\boldsymbol{W}$ in another order.

On this basis, we still follow the Construction. A.56 in the first part. The tight lower bound has already been proved as $f(y, x)$, thus we only need demonstrate that with some choice of $m$, $P(\boldsymbol{W} \mid \boldsymbol{U})$ is left-reversible with the above assumption. In practice, we choose $m = d - 1$. Then the $P(\boldsymbol{W} \mid \boldsymbol{U})$ is reformulated as

$$\begin{bmatrix} \overbrace{\boldsymbol{P_{11}}}^{(d-1)*(d-1)} & \vdots & \overbrace{\boldsymbol{P_{12}}}^{1*1} \\ \underbrace{\boldsymbol{P_{21}}}_{(\mathscr{W}-d+1)*(d-1)} & \vdots & \underbrace{\boldsymbol{P_{22}}}_{(\mathscr{W}-1)*1} \end{bmatrix} := \begin{bmatrix} \sum\limits_{i=1}^{d-1} P(W = w_i \mid y, x)\boldsymbol{E_{(d-1)*(d-1)}} & \vdots & \begin{bmatrix} f(W = w_1 \mid x^c) \\ f(W = w_2 \mid x^c) \\ ... \\ f(W = w_{\mathscr{W}} \mid x^c) \end{bmatrix} \\ \begin{bmatrix} f(W = w_d \mid y, x) \\ f(W = w_{d+1} \mid y, x) \\ ... \\ f(W = w_{\mathscr{W}} \mid y, x) \end{bmatrix} \boldsymbol{J_{1*(d-1)}} & \vdots & \end{bmatrix}. \tag{A.63}$$

We make equivalent denotations:

$$\begin{bmatrix} \overbrace{\boldsymbol{P'_{12}}}^{(d-1)*1} \\ \underbrace{\boldsymbol{P'_{22}}}_{(\mathscr{W}-d+1)*1} \end{bmatrix} := \begin{bmatrix} \overbrace{\boldsymbol{P_{12}}}^{1*1} \\ \underbrace{\boldsymbol{P_{22}}}_{(\mathscr{W}-1)*1} \end{bmatrix} \tag{A.64}$$

In the following part, we claim that we only need to prove $\boldsymbol{P'_{22}} - \boldsymbol{P_{21}}\boldsymbol{P_{11}^{-1}}\boldsymbol{P'_{12}} \neq \boldsymbol{0}$. We do the following algebraic distortion:

$$\begin{bmatrix} \boldsymbol{E_{(d-1)*(d-1)}} & \boldsymbol{0_{(d-1)*(\mathscr{W}-d+1)}} \\ -\boldsymbol{P_{21}}\boldsymbol{P_{11}^{-1}} & \boldsymbol{E_{(\mathscr{W}-d+1)*(\mathscr{W}-d+1)}} \end{bmatrix} * \begin{bmatrix} \boldsymbol{P_{11}} & \boldsymbol{P'_{12}} \\ \boldsymbol{P_{21}} & \boldsymbol{P'_{22}} \end{bmatrix} = \begin{bmatrix} \boldsymbol{P_{11}} & \boldsymbol{P'_{12}} \\ \boldsymbol{0_{(\mathscr{W}-d+1)*(d-1)}} & \boldsymbol{P'_{22}} - \boldsymbol{P_{21}}\boldsymbol{P_{11}^{-1}}\boldsymbol{P'_{12}} \end{bmatrix}. \tag{A.65}$$

According to the well-known Sylvester's inequality [Matsaglia and PH Styan, 1974]: $\forall \boldsymbol{A_{m*n}}, \boldsymbol{B_{n*p}}$, we have $\min\{rank(\boldsymbol{A}), rank(\boldsymbol{B})\} \geq rank(\boldsymbol{AB}) \geq rank(\boldsymbol{A}) + rank(\boldsymbol{B}) - n$. Then we have

$$rank\left( \begin{bmatrix} \boldsymbol{P_{11}} & \boldsymbol{P'_{12}} \\ \boldsymbol{P_{21}} & \boldsymbol{P'_{22}} \end{bmatrix} \right) = rank\left( \begin{bmatrix} \boldsymbol{P_{11}} & \boldsymbol{P'_{12}} \\ \boldsymbol{0_{(\mathscr{W}-d+1)*(d-1)}} & \boldsymbol{P'_{22}} - \boldsymbol{P_{21}}\boldsymbol{P_{11}^{-1}}\boldsymbol{P'_{12}} \end{bmatrix} \right). \tag{A.66}$$

If $P'_{22} - P_{21}P_{11}^{-1}P'_{12} = 0_{(\mathscr{W}-d+1)*(d-1)}$, then the right side of $rank()$ will be equal to $rank(\left[P_{11}, P'_{12}\right]) = d-1 < d$. On the other hand, if $P'_{22} - P_{21}P_{11}^{-1}P'_{12} \neq 0_{(\mathscr{W}-d+1)*(d-1)}$, then it will turn to be $d$ (full column rank). In conclusion, to demonstrate the left-reversibility of $P(W \mid U)$, $P'_{22} - P_{21}P_{11}^{-1}P'_{12} \neq 0$ is all we need.

If we use $[\cdot]_{(i)}$ to denote the $i$-th element of vector $i = d, ...\mathscr{W}$, then

$$[P'_{22} - P_{21}P_{11}^{-1}P'_{12}]_{(i)} = \sum_{i=1}^{d-1} f(W = w_i \mid x^c) \left[ \frac{f(W = w_i, x^c)}{\sum\limits_{i=1}^{d-1} f(W = w_i, x^c)} - \frac{f(y, W = w_i, x)}{\sum\limits_{i=1}^{d-1} f(y, W = w_i, x)} \right]. \tag{A.67}$$

We make the contradiction. If we have $P'_{22} - P_{21}P_{11}^{-1}P'_{12} = 0_{(\mathscr{W}-d+1)*(d-1)}$, then

$$
\begin{aligned}
\|P'_{22} - P_{21}P_{11}^{-1}P'_{12}\|_1 &= \sum_{i=1}^{d-1} f(W = w_i \mid x^c) \left[ \frac{\sum\limits_{i=d}^{\mathscr{W}} f(W = w_i, x^c)}{\sum\limits_{i=1}^{d-1} f(W = w_i, x^c)} - \frac{\sum\limits_{i=d}^{\mathscr{W}} f(y, W = w_i, x)}{\sum\limits_{i=1}^{d-1} f(y, W = w_i, x)} \right] \\
&= \sum_{i=1}^{d-1} f(W = w_i \mid x^c) \left[ \frac{f(x^c)}{\sum\limits_{i=1}^{d-1} f(W = w_i, x^c)} - \frac{f(y, x)}{\sum\limits_{i=1}^{d-1} f(y, W = w_i, x)} \right] \\
&= \sum_{i=1}^{d-1} f(W = w_i \mid x^c) \left[ \frac{1}{\sum\limits_{i=1}^{d-1} f(W = w_i \mid x^c)} - \frac{1}{\sum\limits_{i=1}^{d-1} f(W = w_i \mid y, x)} \right] = 0.
\end{aligned}
\tag{A.68}
$$

Thus we have $\sum\limits_{i=1}^{d-1} f(W = w_i \mid x^c) = \sum\limits_{i=1}^{d-1} f(W = w_i \mid y, x)$. Then we substitute it into Eqn (A.67), we have

$$f(W = w_i \mid x^c) - f(W = w_i \mid x, y) = 0, \forall i \in \{d, ...\mathscr{W}\}. \tag{A.69}$$

Contradiction! Hence we have $P_{22} - P_{21}P_{11}^{-1}P_{12} \neq 0_{(\mathscr{W}-d+1)*(d-1)}$, and then $P(W \mid U)$ in Construction. A.63 is left-reversible. Proved.

- **Conclusion 3:** If $P(W \mid U)$ is restricted to be left-reversible and $f(W \mid \neg x) = f(W \mid x, y)$, then the tight lower bound of $f(Y_x = y)$ is $f(y \mid x)$.

If this assumption holds, we will have $P'_{22} - P_{21}P_{11}^{-1}P'_{12} = 0$ in the above construction, thus $P(W \mid U)$ will be irreversible and validates the condintion here. Hence we need another way.

According to the left-reversibility of $P(W \mid U)$, we have

$$f(U \mid x, y) = P(W \mid U)^{-1}f(W \mid x, y) = P(W \mid U)^{-1}f(W \mid x^c) = f(U \mid x^c) \tag{A.70}$$

Then we have

$$
\begin{aligned}
f(Y_x = y) &= f(x,y) + \sum_{i=1}^{d} \frac{f(x,y,u_i)}{f(x,u_i)} f(u_i, x^c) \\
&= f(x,y) + f(x,y)f(x^c) \sum_{i=1}^{d} \frac{f(u_i \mid x,y)}{f(x,u_i)} f(u_i \mid x^c) \\
&= f(x,y) + f(x,y)f(x^c) \sum_{i=1}^{d} \frac{f(u_i \mid x,y)^2}{f(x,u_i)} \\
&\overset{*}{\geq} f(x,y) + f(x,y)f(x^c) \frac{(\sum_{i=1}^{d} f(u_i \mid x,y))^2}{\sum_{i=1}^{d} f(x,u_i)} \\
&= f(x,y) \left( 1 + \frac{f(x^c)}{f(x)} \right) \\
&= f(y \mid x).
\end{aligned}
\tag{A.71}
$$

According to the Chauchy's inequality, the $'\geq'$ ($*$) turns to be $'='$ if and only if $f(\boldsymbol{U} \mid x,y) = f(\boldsymbol{U} \mid x)$. Combining with Eqn (A.70), we have $f(\boldsymbol{U} \mid x,y) = f(\boldsymbol{U} \mid x) = f(\boldsymbol{U} \mid x^c) = f(\boldsymbol{U})$. It holds if and only if $f(\boldsymbol{W} \mid x,y) = f(\boldsymbol{W} \mid x) = f(\boldsymbol{W} \mid x^c) = f(\boldsymbol{W})$, or else the lower bound is not tight.

**Generalisability** Our PI-SFP approach's generalizability can be highlighted in two ways: 1) PI-SFP can handle cases where either reversibility or total observability, or both, do not exist, which renders the literature on single-proxy control ineffective. 2) Extending PI-SFP to incorporate negative control (as shown in Fig.1(b) and Fig.1(c)) is an optional add-on and not a necessity. This simplicity eliminates the need for numerous assumptions, such as completeness and bridge function, which are present in previous double negative control literature [Miao et al., 2018, Cui et al., 2020, Tchetgen et al., 2020, Deaner, 2018, Shi et al., 2020, Singh, 2020, Nagasawa, 2018, Kallus et al., 2021]. The negative control extension to Fig. 1(b)-1(c) will be discussed in the next subsection.

**Verifiability** The feasibility of Assumption 1 has been suggested in previous work. Kuroki et al. Kuroki and Pearl [2014] suggested that the bounds $\underline{P(\boldsymbol{W} \mid \boldsymbol{U})}$ and $\overline{P(\boldsymbol{W} \mid \boldsymbol{U})}$ can be determined a priori through the Bayesian strategy [Greenland, 2005] and some re-calibration methods [Rothman et al., 2008, Selén, 1986]. In their "Head Start Program," they provide a detailed estimation of $P(\boldsymbol{W} \mid \boldsymbol{U})$ to support this claim.

**Practical correspondence**

- Some general cases (just conduct sampling upon $\boldsymbol{U}$): This hypothesis is commonly encountered in real situations, with Kuroki, Judea Pearl Kuroki and Pearl [2014] (page 4) and Li, Judea Pearl Li and Pearl [2022] specifically illustrating how to sample $U$ to infer approximate/bounded estimates of $P(\boldsymbol{W} \mid \boldsymbol{U})$, and mentioning such sampling method has been previously and commonly used, such as fundamental work Greenland [2005], Rothman et al. [2008], Carroll et al. [2006], Selén [1986].

- Concrete example 1 for $P(\boldsymbol{W} \mid \boldsymbol{U})$ (recommendation system: $\boldsymbol{W}$ denotes the popularity of $\boldsymbol{U}$): There are also some more practical examples in our life. For instance, in the context of recommendation systems, a significant amount of work uses the representation of product-user features as a confounder $\boldsymbol{U}$. However, this representation often includes sensitive information, leading to incomplete observations of $\boldsymbol{U}$. Building upon this, the popularity ranking of products in different regions and time periods is publicly available information, which can be used as a proxy variable $\boldsymbol{W}$ for products. By analyzing the different purchasing tendencies of various demographic groups, we can obtain upper and lower bounds estimates for the transition matrix $P(\boldsymbol{W} \mid \boldsymbol{U})$. In this scenario, firstly, $P(\boldsymbol{W} \mid \boldsymbol{U})$ is often irreversible because popularity ranking information itself can be seen as an indicator/projection, and the information it carries is not as rich as the product features. Secondly, we typically can only estimate the upper and lower bounds of $P(\boldsymbol{W} \mid \boldsymbol{U})$ (through methods like Bayesian estimation), as the sampling estimation process for $U$ is likely to be biased.

- Concrete example 2 for $P(\boldsymbol{W} \mid \boldsymbol{U})$ (privacy protection: $\boldsymbol{W}$ is the de-identified $\boldsymbol{U}$): Our PI-SFP framework is also related to privacy-protecting scenario. Survey collectors often need to gather some sensitive information $\boldsymbol{U}$. To obtain more accurate responses and avoid the risk of disclosing personal privacy, they often ask survey respondents to answer some yes-or-no questions using the Randomized Response method. We consider the survey results as $\boldsymbol{W}$, with the following steps: First, the survey respondent flips a coin (with equal probability of heads or tails), and only they know the result. If it lands heads, they answer the question truthfully; if it lands tails, they flip the coin again (with only them knowing the result); if the second toss is headed, they answer "Yes"; if it is tails, they answer "No". With this setup,

even if we do not know $U$, we can deduce $P(W)$ and $P(\boldsymbol{W}|\boldsymbol{U}) = [3/4, 1/4; 1/4, 3/4]$. Of course, this simple setup may still expose other sensitive information, such as the joint distribution of $P(\boldsymbol{Y}, \boldsymbol{U}, \boldsymbol{X})$, which is something the government would not want to see or make public. Therefore, in practical use, for highly confidential information which requires the strongest privacy protection, we tend to develop a more complex/dynamic/irreversible $P(\boldsymbol{W}|\boldsymbol{U})$ (i.e., privacy-protecting algorithms) and manually set its upper and lower bounds.

### A.7.2 Discussion 2: algorithm comparison and acceleration

In this section, we discuss two additional optimization methods which are potential to solve our partial observability problem. We subsequently illustrate the superior performance of PI-SFP compared to these methods. Furthermore, we introduce a novel pruning strategy supported by a local optimization method, which accelerates the optimization process.

**Algorithm comparison** The author of Shen et al. [2017] derived an $\varepsilon-$approximation method that can be utilized in our problem, and the outcome will lie within $[f(Y_x = y), (1 + \varepsilon)f(Y_x = y)]$. However, this algorithm exhibits an exponential time complexity for the dimension $d_u$, making it challenging to operate effectively in high-dimensional confoundings. Additionally, an iterative algorithm was developed in Le Thi et al. [2014] to search the Karush-Kuhn-Tucker (KKT) point of the difference-in-convex (DC) problem, which can be applied to (3). Nevertheless, KKT theory cannot guarantee global optimality compared to our PI-SFP.

**Algorithm acceleration** To expedite the PI-SFP process, we propose setting sufficient criteria to evaluate whether the current partition contains the optimal solution. If the criteria are not met, we can delete the branch online and narrow our search. To this end, we propose a new auxiliary algorithm specifically designed to search for the local minimum of $f(Y_x = y)$, which serves as an upper-bound of $f(Y_x = y)$. We achieve this by implementing the algorithm in the sub-simplex $S$. Specifically, if the optimal value $f_S(Y_x = y)$ is even larger than the local minimum, then it will be larger than $f(Y_x = y)$. Therefore, we claim that this partition must not include the optimal solutions and can be removed permanently. The auxiliary local optimization algorithm is provided as follows. The principle of our algorithm is based on the lemma:

**lemma 7** $\forall i, j$, *if we make adjustment:*

$$
\begin{bmatrix} \breve{\theta}_i & \breve{\theta}_j \\ \breve{\psi}_i & \breve{\psi}_j \\ \breve{\omega}_i & \breve{\omega}_j \end{bmatrix} = \begin{bmatrix} \theta_i & \theta_j \\ \psi_i & \psi_j \\ \omega_i & \omega_j \end{bmatrix} \begin{bmatrix} \alpha & 1-\alpha \\ 1-\alpha & \alpha \end{bmatrix}, \; where \; \alpha \in \begin{cases} (0,1] \; if \; (\frac{\theta_i}{\theta_j} - \frac{\psi_i}{\psi_j})(\frac{\omega_i}{\omega_j} - \frac{\psi_i}{\psi_j}) \geq 0. \\ [1,+\infty) \; if \; (\frac{\theta_i}{\theta_j} - \frac{\psi_i}{\psi_j})(\frac{\omega_i}{\omega_j} - \frac{\psi_i}{\psi_j}) \leq 0, \end{cases} \tag{A.72}
$$

*Then we have*

$$
\sum_{m=i,j} \frac{\breve{\theta}_m}{\breve{\psi}_m} \breve{\omega}_m \leq \sum_{m=i,j} \frac{\theta_m}{\psi_m} \omega_m. \tag{A.73}
$$

We consider the case $\alpha \in (0,1)$, and the second case is symmetric. Due to $(\frac{\theta_i}{\theta_j} - \frac{\psi_i}{\psi_j})(\frac{\omega_i}{\omega_j} - \frac{\psi_i}{\psi_j}) \geq 0$, we have

$$
(\theta_i - \theta_j)(\omega_j - \omega_i)\psi_i\psi_j + (\theta_j\omega_j\psi_i - \theta_i\omega_i\psi_j)(\psi_j - \psi_i) \leq 0. \tag{A.74}
$$

If we denote that

$$
Q_{ij} := \alpha\psi_i\psi_j(\theta_i - \theta_j)(\omega_j - \omega_i)\left[(1-\alpha)\psi_i + \alpha\psi_j\right] + (\theta_j\omega_j\psi_i - \theta_i\omega_i\psi_j)\left[(1-\alpha)\psi_i + \alpha\psi_j\right]\psi_j. \tag{A.75}
$$

Then Eqn (A.74) is equal to

$$
Q_{ij} \leq -Q_{ji}. \tag{A.76}
$$

Furthermore, we find

$$
Q_{ij} = \psi_j\breve{\psi}_j\left[\alpha\psi_i(\theta_i - \theta_j)(\omega_j - \omega_i) + \theta_j\omega_j\psi_i - \theta_i\omega_i\psi_j\right] = \frac{1}{1-\alpha}\psi_j\breve{\psi}_j\left[\breve{\theta}_i\breve{\omega}_i\psi_i - \theta_i\omega_i\breve{\psi}_i\right]. \tag{A.77}
$$

Hence Eqn (A.76) can be transformed as

$$
\psi_j\breve{\psi}_j\left[\breve{\theta}_i\breve{\omega}_i\psi_i - \theta_i\omega_i\breve{\psi}_i\right] < -\psi_i\breve{\psi}_i\left[\breve{\theta}_j\breve{\omega}_j\psi_j - \theta_j\omega_j\breve{\psi}_j\right] \tag{A.78}
$$

Hence

$$\sum_{m=i,j} \frac{\breve{\theta}_m}{\breve{\psi}_m}\breve{\omega}_m \leq \sum_{m=i,j} \frac{\theta_m}{\psi_m}\omega_m. \tag{A.79}$$

Thus we have proved. By this strategy, we should choose suitable $\alpha$ to satisfy $\phi \in IR_{\Phi}$, namely that $f(y, \boldsymbol{W}, \boldsymbol{U}, \boldsymbol{X}) \in \widetilde{\mathcal{F}}$.

### A.7.3 Discussion 3: Extension to Fig. 1(b) and 1(c)

**Fig. 1(a)** Our algorithm PI-SFP mainly focuses on Fig. 1(a). Moreover, if $\boldsymbol{W} \to \boldsymbol{Y}$ is added, the optimization problem will be transferred under Assumption. 1.

$$\begin{aligned}
&\min f(y \mid \boldsymbol{U}, \boldsymbol{X} = x)f(\boldsymbol{U}), \\
&\text{subject to } f(y \mid \boldsymbol{U}, \boldsymbol{X} = x)f(\boldsymbol{U} \mid \boldsymbol{X} = x) = f(y \mid x), \\
&\text{where } f(\boldsymbol{U} \mid \boldsymbol{X} = x) \text{ satisfies} \\
&\begin{bmatrix} \underline{f(\boldsymbol{W} \mid \boldsymbol{U})}f(\boldsymbol{U} \mid \boldsymbol{X} = x) - f(\boldsymbol{W} \mid \boldsymbol{X} = x) \\ f(\boldsymbol{W} \mid \boldsymbol{X} = x) - \underline{f(\boldsymbol{W} \mid \boldsymbol{U})}f(\boldsymbol{U} \mid \boldsymbol{X} = x) \end{bmatrix} \geq 0, \\
&\begin{bmatrix} \underline{f(\boldsymbol{W} \mid \boldsymbol{U})}f(\boldsymbol{U}) - f(\boldsymbol{W}) \\ f(\boldsymbol{W}) - \underline{f(\boldsymbol{W} \mid \boldsymbol{U})}f(\boldsymbol{U}) \end{bmatrix} \geq 0.
\end{aligned} \tag{A.80}$$

Notice that the feasible region of $f(y \mid \boldsymbol{U}, \boldsymbol{X} = x)$ and $f(\boldsymbol{U})$ is even more irregular than in (3). Nevertheless, we can still adopt an analogous strategy to PI-SFP to approximate its optimal value. This remains a topic for future research.

**Fig. 1(b) and 1(c)** The double negative control via introducing auxiliary exposure $\boldsymbol{Z}$ can enhance our estimation. Due to the fact $f(y \mid u, x) = f(y \mid u, x, Z)$, we have:

$$\underline{f(Y_x = y)} := f(y, x) + \max_{\mathcal{Z} \subseteq \boldsymbol{Z}}$$

$$\min_{f(y, \boldsymbol{W}, \boldsymbol{U}, \boldsymbol{X}, \mathcal{Z}) \in \widetilde{\mathcal{F}}_{\mathcal{Z}}} \sum_{u=1}^{d} \frac{f(y, u_i, x, z \in \mathcal{Z})f(u_i, \neg x)}{f(u_i, x, z \in \mathcal{Z})}. \tag{A.81}$$

The feasible region $\widetilde{\mathcal{F}}_{\mathcal{Z}}$ of $f(y, \boldsymbol{W}, \boldsymbol{U}, \boldsymbol{X}, \mathcal{Z})$ is constructed analogously to that of $\widetilde{\mathcal{F}}$. Specifically, for each subset $\mathcal{Z} \subseteq Z$, we can apply PI-SFP and select the maximum of them as the best lower bound of $\underline{f(Y_x = y)}$. We will conduct a more detailed analysis in our subsequent work, particularly in comparison to the performance of the single-proxy control.

$\widetilde{\mathcal{F}}_Z$ is identified as follows:

We denote

$$\begin{aligned}
\theta_i &= f(y, u_i, x, z \in \mathcal{Z}) & \boldsymbol{\theta}_{\mathcal{Z}} &= (\theta_1, \theta_2, ...\theta_d)^T \\
\psi_i &= f(u_i, x, z \in \mathcal{Z}) &, \boldsymbol{\psi}_{\mathcal{Z}} &= (\psi_1, \psi_2, ...\psi_d)^T, \ \boldsymbol{\phi}_{\mathcal{Z}} = (\boldsymbol{\theta}_{\mathcal{Z}}\ \boldsymbol{\psi}_{\mathcal{Z}}\ \boldsymbol{\omega}_{\mathcal{Z}}). \\
\omega_i &= f(u_i, \neg x) & \boldsymbol{\omega}_{\mathcal{Z}} &= (\omega_1, \omega_2, ...\omega_d)^T
\end{aligned} \tag{A.82}$$

where $f(y, \boldsymbol{W}, \boldsymbol{U}, \boldsymbol{X}, \mathcal{Z}) \in \widetilde{\mathcal{F}}_Z = \{\boldsymbol{\phi}_{\mathcal{Z}} \in IR_Z, IR_Z = IR_Z^1 \cap IR_Z^2\}$ leads to the following constraints that we really use:

$$IR_Z^1 = \left\{\boldsymbol{\phi}_Z : \begin{bmatrix} -\boldsymbol{I}_{d*d} \\ \boldsymbol{I}_{d*d} \end{bmatrix} \begin{bmatrix} f(y, \boldsymbol{W}, x, z \in \mathcal{Z})^T \\ f(\boldsymbol{W}, x, z \in \mathcal{Z})^T \\ f(\boldsymbol{W}, \neg x, z \in \mathcal{Z})^T \end{bmatrix}^T - \begin{bmatrix} -\overline{P(\boldsymbol{W} \mid \boldsymbol{U})} \\ \underline{P(\boldsymbol{W} \mid \boldsymbol{U})} \end{bmatrix} \boldsymbol{\phi}_{\mathcal{Z}} \geq \boldsymbol{0}\right\}. \tag{A.83}$$

Moreover, the set $IR_{\Phi}^2$ indicates the natural constraints by default:

$$IR_Z^2 = \left\{\boldsymbol{\phi}_Z : \begin{bmatrix} \boldsymbol{1}^T\boldsymbol{\theta} \\ \boldsymbol{1}^T\boldsymbol{\phi} \\ \boldsymbol{1}^T\boldsymbol{\omega} \end{bmatrix} = \begin{bmatrix} f(y, x, z \in \mathcal{Z}) \\ f(x, z \in \mathcal{Z}) \\ f(\neg x, z \in \mathcal{Z}) \end{bmatrix}, \forall i, \begin{cases} \theta_i \in [0, f(y, x, z \in \mathcal{Z})] \\ \phi_i \in (0, f(x, z \in \mathcal{Z})] \\ \omega_i \in [0, f(\neg x, z \in \mathcal{Z})] \end{cases}\right\}. \tag{A.84}$$

### A.7.4 Discussion 4: the continuous confoundings

**Assumption 3** *(partial observability assumption for continuous confoundings)* $P(\boldsymbol{W} \mid \boldsymbol{U}) \in \mathscr{P}$, *where* $\mathscr{P} =$
$\left\{P(\boldsymbol{W} \mid \boldsymbol{U}) : \begin{bmatrix} P(\boldsymbol{W} \mid \boldsymbol{U} \in [u_{i-1}, u_i]) - P(\boldsymbol{W} \mid \boldsymbol{U} \in [u_{i-1}, u_i]) \\ P(\boldsymbol{W} \mid \boldsymbol{U} \in [u_{i-1}, u_i]) - \underline{P(\boldsymbol{W} \mid \boldsymbol{U} \in [u_{i-1}, u_i])} \end{bmatrix}$ *is non-negative*$\right\}$.

**Assumption 4** *(Lipschitz condition)* $\forall y \in \boldsymbol{Y}, \forall \{u', u''\} \in \boldsymbol{U}$, *we have* $\left| \frac{f(y,u',x)-f(y,u'',x)}{f(u',x)-f(u'',x)} \right| \leq C_1, \left| \frac{f(u',x)-f(u'',x)}{u'-u''} \right| \leq$
$C_2$, *where* $C_1, C_2$ *are positive constants.*

**lemma 8** *Suppose that Assumption. 3-4 hold.* $\forall i \in \{0, 1, ...d - 1\}, \forall u \in [u_i, u_{i+1}]$, *we have*

$$\frac{\int_{u_i}^{u_{i+1}} f(y,u,x)du}{\int_{u_i}^{u_{i+1}} f(u,x)du} \leq \frac{f(y,u,x)}{f(u,x)} \frac{1}{1-\frac{1}{2}C_2\eta} + \frac{\frac{1}{2}C_1C_2\eta}{1-\frac{1}{2}C_2\eta}. \tag{A.85}$$

*On the other hand,*

$$\frac{\int_{u_i}^{u_{i+1}} f(y,u,x)du}{\int_{u_i}^{u_{i+1}} f(u,x)du} \geq \frac{f(y,u,x)}{f(u,x)} \frac{1}{1+\frac{1}{2}C_2\eta} - \frac{\frac{1}{2}C_1C_2\eta}{1+\frac{1}{2}C_2\eta}. \tag{A.86}$$

**The proof of lemma.** (8) We do partition on the confounding interval $[U^L, U^U]$ as $[u_0, u_1, u_2, ..., u_{d-1}, u_d]$, where $u_0 = U^L, u_d = U^U$. The independent variables is re-defined by

$$\theta_i = f(y, U \in [u_i, u_{i+1}], x), \psi_i = f(U \in [u_i, u_{i+1}], x), \omega_i = f(U \in [u_i, u_{i+1}], \neg x), \tag{A.87}$$

$$\begin{aligned} \forall u' \in [u_i, u_{i+1}], \frac{\int_{u_i}^{u_{i+1}} f(y,u,x)du}{\int_{u_i}^{u_{i+1}} f(u,x)du} &\leq \frac{\int_{u_i}^{u_{i+1}} \left[ f(y,u',x) + C_1 \left| f(u,x) - f(u',x) \right| \right] du}{\int_{u_i}^{u_{i+1}} \left[ f(u',x) + (f(u,x) - f(u',x)) \right] du} \\ &\leq \frac{f(y,u',x)(u_{i+1} - u_i) + C_1C_2\frac{1}{2}(u_{i+1}-u_i)^2}{f(u',x)(u_{i+1}-u_i) - C_2\frac{1}{2}(u_{i+1}-u_i)^2} \\ &\leq \frac{\frac{f(y,u',x)}{f(u',x)} + \frac{\frac{1}{2}C_1C_2\eta\delta}{f(u',x)}}{1 - \frac{\frac{1}{2}C_2\eta\delta}{f(u',x)}} \\ &\leq \frac{f(y,u',x)}{f(u',x)} \frac{1}{1-\frac{1}{2}C_2\eta} + \frac{\frac{1}{2}C_1C_2\eta}{1-\frac{1}{2}C_2\eta}. \end{aligned} \tag{A.88}$$

Here $i = 0, 1, ...d - 1$. Analogously, we can prove the other direction. Thus we have proved the lemma.

**Corollary 1** *(PI-SFP's error for continuous confoundings) Suppose that Assumption. 3-4 holds. When $U$ is continuous, and* $\max\limits_{i \in \{1,2,...d\}} |u_i - u_{i-1}| < \eta\delta$. *Then* $\underline{f(Y_x = y)} \leq \lim\limits_{n \to +\infty} \underline{f_{opt}^n(Y_x = y)} \leq \frac{1}{1-\frac{1}{2}C_2\eta} \underline{f(Y_x = y)} + \frac{C_1 f(\neg x) - f(y,x)}{2-C_2\eta} C_2\eta$.

Then we prove this corollary. **The proof of Corollary. 1** If we use $\underline{f(y, \boldsymbol{U}, x)}, \underline{f(\boldsymbol{U}, x)}, \underline{f(\boldsymbol{U}, \neg x)}$ to denote the optimal

solution of the optimal value $f(Y_x = y)$ in the continuous case, then we have

$$
\begin{aligned}
&f(Y_x = y) - f(y, x) \\
&= \int_{U^L}^{U^U} \frac{f(y, u, x)}{f(u, x)} f(u, \neg x) du \\
&= \sum_{i=0}^{d-1} \int_{u_i}^{u_{i+1}} \frac{f(y, u, x)}{f(u, x)} f(u, \neg x) du \\
&\geq \sum_{i=0}^{d-1} \int_{u_i}^{u_{i+1}} \left[ \frac{\int_{u_i}^{u_{i+1}} f(y, u, x) du}{\int_{u_i}^{u_{i+1}} f(u, x) du} - \frac{\frac{1}{2} C_1 C_2 \eta}{1 - \frac{1}{2} C_2 \eta} \right] \left( 1 - \frac{1}{2} C_2 \eta \right) f(u, \neg x) du \\
&= \left( 1 - \frac{1}{2} C_2 \eta \right) \sum_{i=0}^{d-1} \frac{\int_{u_i}^{u_{i+1}} f(y, u, x) du}{\int_{u_i}^{u_{i+1}} f(u, x) du} \int_{u_i}^{u_{i+1}} f(u, \neg x) du - \frac{1}{2} C_1 C_2 \eta f(\neg x).
\end{aligned}
$$

$$(A.89)$$

Here $\{\int_{u_i}^{u_{i+1}} f(y, u, x) du, \int_{u_i}^{u_{i+1}} f(u, x) du, \int_{u_i}^{u_{i+1}} f(u, \neg x) du, i = 0, 1, ...d - 1\}$ is within the feasible region of PI-SFP in the discrete case. Then we have

$$
(A.89) \geq \left( 1 - \frac{1}{2} C_2 \eta \right) \left( \lim_{n \to +\infty} f^n_{opt}(Y_x = y) - f(y, x) \right) - \frac{1}{2} C_1 C_2 \eta f(\neg x)
$$

$$
f(Y_x = y) \geq \left( 1 - \frac{1}{2} C_2 \eta \right) \lim_{n \to +\infty} f^n_{opt}(Y_x = y) + \frac{1}{2} C_2 \eta f(y, x) - \frac{1}{2} C_1 C_2 \eta f(\neg x)
$$

$$
\lim_{n \to +\infty} f^n_{opt}(Y_x = y) \leq \frac{1}{1 - \frac{1}{2} C_2 \eta} f(Y_x = y) + \frac{\frac{1}{2} C_1 f(\neg x) - \frac{1}{2} f(y, x)}{1 - \frac{1}{2} C_2 \eta} C_2 \eta
$$

$$(A.90)$$

On the other hand, each optimal solution by PI-SFP corresponds to a solution in the continuous case. Namely if the discrete PI-SFP's optimal solution is denoted as $\{\int_{u_i}^{u_{i+1}} f(y, u, x) du, \int_{u_i}^{u_{i+1}} f(u, x) du, \int_{u_i}^{u_{i+1}} f(y, u, \neg x) du, i = 0, 1, ...d - 1\}$. Then we can construct

$$
\begin{aligned}
f^{opt}(y, u, x) &= \frac{\int_{u_i}^{u_{i+1}} f(y, u, x) du}{u_{i+1} - u_i}, u \in [u_i, u_{i+1}). \\
f^{opt}(u, x) &= \frac{\int_{u_i}^{u_{i+1}} f(u, x) du}{u_{i+1} - u_i}, u \in [u_i, u_{i+1}). \\
f^{opt}(u, \neg x) &= \frac{\int_{u_i}^{u_{i+1}} f(u, \neg x) du}{u_{i+1} - u_i}, u \in [u_i, u_{i+1})
\end{aligned}
$$

$$(A.91)$$

as one of the solution in the continuous case. Hence $\lim_{n \to +\infty} f^n_{opt}(Y_x = y) \geq f(Y_x = y)$. We have finished the proof.

### A.7.5 Comment on Theorem 1: the relationship between $L_n$ and $n$

**Remark 2** (*Hardness of establishing functional associations between $L_n$ and $n$*) *The worst-case scenario for $L_n$ is $L_n = \lfloor log(n) \rfloor + 1$. In this situation, PI-SFP is equivalent to the method of exhaustion, which exhibits slow polynomial convergence, as shown in Theorem 1, with a rate of $O(n^{-\alpha})$, where $\alpha = \frac{1}{4d} log(\frac{2}{\sqrt{3}})$. However, empirical evidence suggests that this scenario is rare, and in the simulation section, the convergence rate is faster than $O(n^{-\alpha})$. Additionally, pruning strategies can be employed to further improve the convergence rate, which is discussed in Section. 7.*

*Despite this empirical observation, it is well beyond the scope of this paper to theoretically estimate $L_n$ w.r.t $n$. During iteration, each optimal solution (converging point) may be covered by increasing number of nested sequences[11]. These sequences possess different lengths and are difficult to estimate. More seriously, the number of optimal solutions is not necessarily finite either, namely $|\mathbf{\Phi}_{opt}| < +\infty$ may not be guaranteed.*

---

[11]Notice that it is equivalent to guarantee each converging point is covered by finite partitions. It resorts to the regularity condition of simplices (identified in [Ciarlet, 2002]). However, whether Longest-edge bisection can promise a family of regular partitions is still a conjecture [Korotov et al., 2016] to be solved.

Notice that $L_n = O(n)$ when each $\phi_{opt}$ is partitioned via finite number of simplices and $|\phi_{opt}| < +\infty$ in the infinite process. By this motivation, we aim to prove that such finiteness is true under a fairly broad assumption mentioned in Ciarlet [2002], Korotov et al. [2016]. In fact, we address this conjecture and it is our by-contribution:

*what is the maximum intersection number of regular simplicial partitions? In other words, for arbitrary point in $\mathcal{S}_0$, what is the maximum number of simplices it can be affiliated with during partitioning, under fairly broad assumption?*

**Assumption 5**
$$vol(S) \geq \eta h(S)^d. \tag{A.92}$$

*Here $vol(S)$ denotes the volumn of simplex $S$, $h(S)$ denotes the longest edge of $S$, and $d \geq 2$ is the dimension.*

Roughly speaking, the regularity assumption guarantees the simplex would not degenerate to the hyper-plane, or else $\lim\limits_{vol(S) \to 0} \frac{vol(S)}{h(S)^d} = 0$. In other words, $\frac{vol(S)}{h(S)^d}$ can be higher when the simplex "seems to be regular"", namely each edge keeps the same length. In addition, the simplicial partitoins during partitioning are denoted as $\mathcal{S}_0, \mathcal{S}_1, \mathcal{S}_2$.... Our problem can be summarized as follows:

**Theorem 3** *Suppose that Ass. 5 holds. Each point in $\mathcal{S}_0$ is included within at most $\frac{1}{\eta} \left( \frac{2e\pi}{d} \right)^{\frac{d}{2}}$ simplices.*

**proof** For $\gamma_0 \in \mathcal{S}_0$, we construct a ball $\mathcal{B}(\gamma_0, r)$. We use $A(\cdot)$ to denote the surface area of sphere:

$$A(\mathcal{B}(\gamma_0, r) \cap \mathcal{S}_0) = \sum_{S \in \mathcal{S}_k, \gamma_0 \in S} A(\mathcal{B}(\gamma_0, r) \cap S). \tag{A.93}$$

On the one hand, the LHS of Eqn. A.93 can be upper bounded as:

$$A(\mathcal{B}(\gamma_0, r) \cap \mathcal{S}_0) \leq A(\mathcal{B}(\gamma_0, r)) = \frac{2\pi^{\frac{d}{2}}}{\Gamma\left(\frac{d}{2}\right)} r^{d-1} < +\infty. \tag{A.94}$$

On the other hand, we calculate the RHS of Eqn. A.93. To solve it, we take advantage of the following integral (we take $\gamma_0$ as the origin):

$$\int_{\mathcal{W}_S\gamma \geq 0} e^{-\|\gamma\|^2} d\gamma. \tag{A.95}$$

Here for each term in the right side, the sphere of the ball is cut by certain facets of each S, whose normal vector is denoted as a set $\mathcal{W}_S$, whose each row denotes a $1 * 4d$ normal vector. Without loss of generation, for any facets, we assume that its normal vector points to the remaining supporting vector, namely their inner product is positive.

Here $\|\cdot\|$ also denotes the Euclidean norm. If we use the polar coordinates, we can get a new expression by differential element method:

$$\int_{\mathcal{W}_S\gamma \geq 0} e^{-\|\gamma\|^2} d\gamma = \int_{\mathcal{W}_S\gamma \geq 0} d\Omega \int_0^{+\infty} e^{-l^2} l^{d-1} dl = \frac{A(\mathcal{B}(\gamma, r) \cap S)}{r^{d-1}} \int_0^{+\infty} e^{-l^2} l^{d-1} dl. \tag{A.96}$$

According to Eqn. A.94-A.96, we have

$$A(\mathcal{B}(\gamma_0, r) \cap S) = \frac{2 \int_{\mathcal{W}_S\gamma \geq 0} e^{-\|\gamma\|^2} d\gamma}{\Gamma\left(\frac{d}{2}\right)} r^{d-1}$$
$$\forall k, \frac{A(\mathcal{B}(\gamma_0, r) \cap S)}{A(\mathcal{B}(\gamma_0, r) \cap \mathcal{S}_0)} \geq \frac{\int_{\mathcal{W}_S\gamma \geq 0} e^{-\|\gamma\|^2} d\gamma}{\pi^{\frac{d}{2}}}. \tag{A.97}$$

Hence we only need prove the integral is lower bounded by a constant above zero. We will do this by extracting a sub-space from $\mathcal{W}_S\gamma \geq 0$, which is easy to be integrated. Specifically, we consider a sub space which is an affine transformation on $\mathcal{W}_S\gamma \geq 0$. We introduce the diameter of simplex $S$ as $\text{dia}(S) = \max_{s_1, s_2 \in S} \|s_1 - s_2\|$.

$$\int_{\mathcal{W}_S \gamma \geq 0} e^{-\|\gamma\|^2} d\gamma \geq \int_{\gamma=t\gamma', \gamma' \in S} e^{-\|\gamma\|^2} d\gamma \quad (\forall t > 0, t \text{ is arbitrarily chosen})$$

$$= \int_{\gamma' \in S} t^{4d} e^{-t^2 \|\gamma'\|^2} d\gamma' \tag{A.98}$$

$$\overset{*}{\geq} t^* \mathrm{Vol}(S) e^{-t^2 dia(S)^2}$$

$$\overset{**}{\geq} \eta t^{4d} h(S)^{4d} e^{-t^2 dia(S)^2}$$

$*$ is due to $\forall \gamma' \in S$, we have $\|\gamma'\| \leq \mathrm{dia}(S)$, since $\gamma_l$ is chosen as the origin. $**$ is due to Ass. 5. Additionally, we further show that in the above Formulation, $h(S) = \mathrm{dia}(S)$. Namely the longest edge of simplex always serves as the diameter. It is equal to prove ($S^i$ denotes the supporting vector):

$$\mathrm{dia}(S) = \max_{s_1, s_2 \in S} \|s_1 - s_2\| \quad \left( s_2 = \sum_i \lambda_i S^i, \lambda_i \in [0,1] \right)$$

$$= \max_{s_1, s_2 \in S} \left\| \left( \sum_i \lambda_i \right) s_1 - \sum_i (\lambda_i S^i) \right\|$$

$$= \max_{s_1, s_2 \in S} \left\| \sum_i \lambda_i (s_1 - S^i) \right\| \tag{A.99}$$

$$\leq \max_{s_1 \in S} \max_i \|s_1 - S^i\|$$

$$\overset{*}{\leq} \max_{i,j} \|S^j - S^i\| \leq h(S).$$

$*$ is due to we also do the expansion on $s_1$, namely $s_1 = \sum_i \lambda_i' S^i, \lambda_i' \in [0,1]$. On the other hand, we have $h(S) \leq \max_{s_1, s_2 \in S} \|s_1 - s_2\| = \mathrm{dia}(S)$ by definition. Thus $\mathrm{dia}(S) = h(S)$. Hence

$$\forall t, \int_{\mathcal{W}_S \gamma \geq 0} e^{-\|\gamma\|^2} d\gamma \geq \eta (t dia(S))^d e^{-\left(t \, dia(S)^2\right)}. \tag{A.100}$$

Due to the arbitrary of $t$, we have

$$\int_{\mathcal{W}_S \gamma \geq 0} e^{-\|\gamma\|^2} d\gamma \geq \eta \max_x x^d e^{-x^2} = \eta x^d e^{-x^2} \Big|_{x=\sqrt{\frac{d}{2}}} = \eta \left(\frac{d}{2}\right)^{\frac{d}{2}} e^{-\frac{d}{2}}. \tag{A.101}$$

Finally, we have

$$\forall k, \frac{A\left(\mathcal{B}\left(\gamma, r\right) \cap S\right)}{A\left(\mathcal{B}\left(\gamma, r\right) \cap S_0\right)} \geq \eta \left(\frac{d}{2e\pi}\right)^{\frac{d}{2}}, \text{ the intersection number } N \leq \frac{1}{\eta} \left(\frac{2e\pi}{d}\right)^{\frac{d}{2}} < +\infty. \tag{A.102}$$

We have finished the proof.

## A.8 SIMULATIONS AND AUXILIARY EXPERIMENTS

**Simulations**    The visualization of our PI-SFP's performance in simulations are presented in Figure 2 and Figure 3.

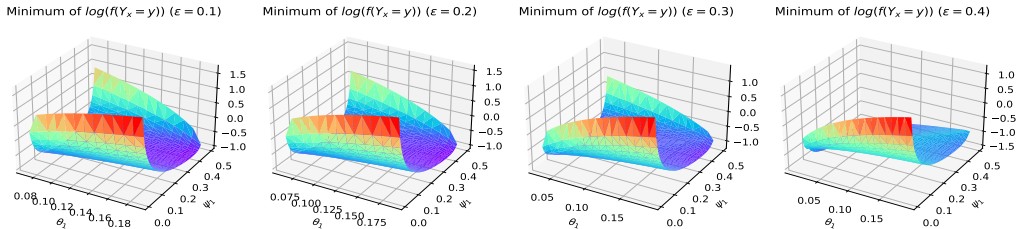

FIGURE 2: *We search the minimum of $f(Y_x = y)$ in the binary case, by conducting naive linear programming on each fixed $\theta_1(\theta_2)$ and $\psi_1(\psi_2)$ in Eqn (3).*

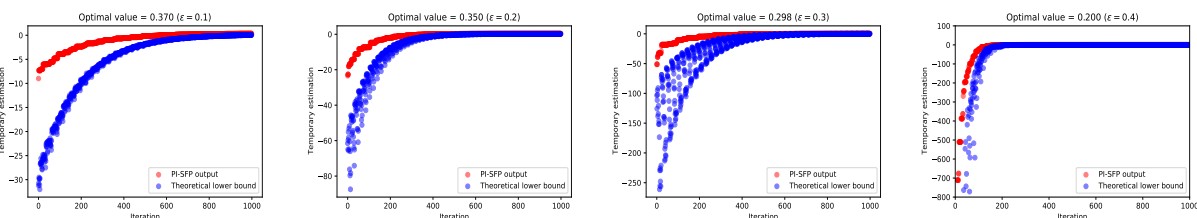

FIGURE 3: *Results of PI-SFP. PI-SFP (blue) converges to the optimal value of $\underline{f(Y_x = y)}$ with $\varepsilon$ changing from $0.1$ to $0.4$. The red line denotes the theoretical convergence rate (ground truth).*

**Real-world experiments**    We present the numerical result of real-world experiments as follows. Here semi-parametric COCA and Doubly-robust parametric COCA method is followed by Tchetgen et al. [2023], Park and Tchetgen [2023]; moreover, the standard difference-in-difference (DID) method is followed by Card and Krueger [1993], Angrist and Pischke [2009].

To facilitate fair comparison, we adopt the same experimental setting as in Tchetgen et al. [2023] and we refer readers to specific details in their experimental part. Specifically, in the dataset, we set $185$ and $488$ municipalities as samples from two areas Pernambuco (PE),Rio Grande do Sul (RS), as the treatment and control group, respectively. The covariate $U$ contain three parts: (i) municipality-level population size, (ii) population density, and (iii) proportion of females measured in 2014. Moreover, we force "post-epidemic municipality-level birth rate" in 2016 as the outcome $Y$, and whether individuals are infected by the virus as treatment control $X$. Finally, we choose "preepidemic municipality-level birth rates in 2013 and 2014" as the outcome proxies $W_1, W_2$, respectively, which is so-called NCO in Table 3. Our transition matrix $P(W \mid U)$ is also approximated from the observations in the public data.

In this process, we choose $P(X, Y), P(W)$ and partial observed transition matrix $P(W \mid U)$ as observed data, and others as the protected feature. Our PI-SFP exhibits a narrower lower and upper bound compared with the previous literature in most cases.

| Estimator | Statistic | NCO | | |
|---|---|---|---|---|
| | | $W_1$ | $W_2$ | $(W_1, W_2)$ |
| Semi-parametric COCA | Estimate | $-2.410$ | $-2.182$ | $-2.180$ |
| | SE | $0.356$ | $0.503$ | $0.342$ |
| | 95%CI | $(-3.107, -1.713)$ | $(-3.168, -1.196)$ | $(-2.850, -1.510)$ |
| Doubly-robust parametric COCA | Estimate | $-2.235$ | $-1.833$ | $-2.182$ |
| | SE | $0.502$ | $0.519$ | $0.415$ |
| | 95%CI | $(-3.220, -1.250)$ | $(-2.850, -0.816)$ | $(-2.996, -1.368)$ |
| Standard DiD | Estimate | $-1.156$ | $-1.041$ | $-1.041$ |
| | SE | $0.199$ | $0.195$ | $0.195$ |
| | 95% CI | $(-1.546, -0.767)$ | $(-1.424, -0.658)$ | $(-1.424, -0.658)$ |
| PI-SFP (ours) | bound | $[-3.012, -1.201]$ | $[-2.732, -1.232]$ | $[-2.742, -1.203]$ |

TABLE 3: *Real-world experiment.*