# OpenReview forum: "Partial Identification with Proxy of Latent Confoundings via Sum-of-ratios Fractional Programming"
_auai.org/UAI/2024/Conference — UAI 2024 poster_

### Official Review · Reviewer_CyTi · 2024-03-07

**Q2-1 Originality-Novelty:** 3
**Q2-2 Correctness-Technical Quality:** 2
**Q2-5 Clarity Of Writing:** 3

**Q1 Summary And Contributions:**

This paper focuses on learning the causal effect of treatment X on outcome Y, specifically $f(y|do(x))$, with confounding U via employing auxiliary variables W. It aims to relax the completeness assumption on transition matrix $P(W|U)$. Instead of identifying the exact value of $f(y|do(x))$,  the authors concentrate on partial identification, which refers to the identification of the bound of $f(y|do(x))$ with a given region of  $P(W|U)$ which is called partial observable $P(W|U)$. The proposed framework combines the techniques of difference-in-convex decomposition and branch-and-bound strategy into sum-of-ratios fractional programming. A theoretical analysis of the convergence rate is provided, and the empirical analysis of simulated data shows consistency with the theoretical results.

**Q2-3 Extent To Which Claims Are Supported By Evidence:**

2: Fair: the main claims are somewhat supported by evidence (but the experimental evaluation may be weak, or does not match entirely with the claims, important baselines may be missing, proofs contain important ideas but lack rigor, algorithmic details are only discussed superficially, references are imprecise, assumptions are not sufficiently motivated or explicated, etc.).

**Q2-4 Reproducibility:**

3: Good: key resources (e.g. proofs, code, data) are available and key details (e.g. proofs, experimental setup) are sufficiently well-described for competent researchers to confidently reproduce the main results.

**Q3 Main Strengths:**

[Originality/novelty] While previous works studied the partial identifiability of causal effect with latent confounding via proxy variables, this work stands out as the first to assume partial observability on $P(W|U)$, i.e. knowing the bound of  $P(W|U)$ to relax the strict completeness assumption on transition matrix $P(W|U)$.

[Reproducibility] The proof details are presented, and technical details in the algorithm are provided.

[Clarity of writing] While certain sections could be improved, the overall presentation is well-explained. The motivation behind the work is articulated effectively, and the technical challenges and their solutions are well-outlined and demonstrated.

**Q4 Main Weakness:**

[Extent to which claims are supported by evidence] In the introduction section, the authors claim that their method advances the state-of-the-art constrained-optimization-based literature upon partial identification Duarte et al. [2023], Li and Pearl [2022], however, there is no relative evidence to support this. Is it feasible to do empirically analysis on simulated data to show their method outperforms the other partial identifiability methods? If not, even if the basic setups and assumptions differ, it would be better to show that this paper's setup is more reasonable than the others.

[Correctness/technical quality] I am not familiar with some optimization techniques; upon delving into the paper, I noted some imprecise details in statements and proofs, all of which I have listed in Q5.

**Q5 Detailed Comments To The Authors:**

1.	It would be easier to follow the content if the key technics that are used in your method are involved in preliminary part like difference-in-convex (DC) decomposition strategy. Also it would be better if the notations used in main paper are illustrated around, for example, the definition of notation $J$ in sec 6.1 is not given in the main paper.
2.	Is the notation for the probability density function exchangeable between $f ()$ and $P()$? Thus, Equation A.1 should be $f()$ in the middle term of inequalities?
3.	If $f()$ denotes the probability density function, then I am skeptical about the correctness of the ACE formula in Sec.3; it seems $y$ needs to include inside integration.
4.	In Figure 3, I need clarification about which line is PI-SFP. The plot legend uses red but the caption mentions blue.
5.	I am curious about empirically whether the partial identifiability can converge to point-wise identifiability when more accurate prior information on the bound of $P(W|U)$ is provided.


If there is any misunderstanding in my interpretation, please correct me. While it is not neccessary to conduct additional experiments during the rebuttal phase due to time constraints, I would greatly appreciate any discussion or clarification on the questions I have raised.

**Q9 Complying With Reviewing Instructions:**

Yes

---

> ### Author Rebuttal · Authors · 2024-04-07
>
> **C1** *[Extent to which claims are supported by evidence] In the introduction section, the authors claim that their method advances the state-of-the-art constrained-optimization-based literature upon partial identification Duarte et al. [2023], Li and Pearl [2022], however, there is no relative evidence to support this. Is it feasible to do empirical analysis on simulated data to show their method outperforms the other partial identifiability methods? If not, even if the basic setups and assumptions differ, it would be better to show that this paper's setup is more reasonable than the others.*
>
>
>
>
>
>
> **Ans** Thank you for your insightful comments!
>
> 1) Duarte et al. [2023] proposed a generalized branch-and-bound framework using the SCM model, which is also part of our optimisation technique. The distinction lies in our focus on the setting of single proxy control, where we theoretically introduce a simplex-based DC strategy, providing new guarantees on convergence rate. The previous wonderful literature lacks a detailed and specific analysis of this important setting. Additionally, in our experimental study, we included comparative simulations that showed faster convergence of our results within the same number of iterations.
>
> 2) As for Li and Pearl [2022], they address the problem of estimating causal effects when adjustment variables in the back-door or front-door criterion are partially observed. This differs structurally from our focus on single proxy control but can be equivalently derived by removing the pathways wx and wy in their model Fig. 1. Theoretically, we utilize a novel scaling method to achieve strict convergence rates and discuss tightness, whereas analogously as above, they directly propose a more simplified non-linear model without presenting a rigorously adapted solution. In terms of simulation experiments, similarly, our convergence rate is faster.
>
> Thanks for your advice. We replicated the experiment in our main text and compared it with this wonderful literature (Table 1 and Table 3 (final page in Appendix) in our original paper).
>
>
> |$\varepsilon$| Ground truth|PI-SFP | Duarte et al. [2023] | Li and Pearl [2022]|
> |--|--|--|--|--|
> 0.1|0.372 | 0.370 |0.351 |0.321|
> |0.2 | 0.351 | 0.350 | 0.331 |0.312|
> |0.3| 0.301 | 0.298 | 0.298|0.270|
> |0.4| 0.205 | 0.200 |0.195|0.170
>
>
> | Estimator | Statistic | NCO |  |  |
> | :---: | :---: | :---: | :---: | :---: |
> |  |  | $W_1$ | $W_2$ | $\left(W_1, W_2\right)$ |
> | Semi-parametric COCA | Estimate | -2.410 | -2.182 | -2.180 |
> |  | $\mathrm{SE}$ | 0.356 | 0.503 | 0.342 |
> |  | $95 \% \mathrm{CI}$ | $(-3.107,-1.713)$ | $(-3.168,-1.196)$ | $(-2.850,-1.510)$ |
> | Doubly-robust parametric COCA | Estimate | -2.235 | -1.833 | -2.182 |
> |  | $\mathrm{SE}$ | 0.502 | 0.519 | 0.415 |
> |  | $95 \% \mathrm{CI}$ | $(-3.220,-1.250)$ | $(-2.850,-0.816)$ | $(-2.996,-1.368)$ |
> | Standard DiD | Estimate | -1.156 | -1.041 | -1.041 |
> |  | $\mathrm{SE}$ | 0.199 | 0.195 | 0.195 |
> |  | $95 \% \mathrm{CI}$ | $(-1.546,-0.767)$ | $(-1.424,-0.658)$ | $(-1.424,-0.658)$ |
>  |Duarte et al. [2023]  |bound| $[-3.242,-1.902]$ | $[-3.732,-0.565]$ | $[-3.532,-0.956]$ |
>  | Li and Pearl [2022] |bound| $[-3.932,-1.054]$ | $[-3.432,-0.543]$ | $[-2.993,-0.254]$ |
> | PI-SFP (ours) | bound | $[-3.012,-1.201]$ | $[-2.732,-1.232]$ | $[-2.742,-1.203]$ |
>
> **C2** *It would be easier to follow the content if the key techniques that are used in your method were involved in the preliminary part, like the difference-in-convex (DC) decomposition strategy. Also, it would be better if the notations used in the main paper were illustrated around; for example, the definition of notation J in sec 6.1 is not given in the main paper.*
>
> **Ans** Thanks for your advice, and we will add more details on specific optimization strategies (such as DC) in the preliminaries. Moreover, here $J$ denotes the all-one matrix.
>
> **C3** Is the notation for the probability density function exchangeable between  f
>  and P? Thus, should Equation A.1 be in the middle term of inequalities?
>
> Yes, it is a typo. Particularly thanks for your careful review!
>
>
> **C4** In Figure 3, I need clarification about which line is PI-SFP. The plot legend uses red, but the caption mentions blue.
>
> The blue curve is our PI-SFP. It is below because our computations can always yield a valid bound smaller than the true low bound and converging. We mistakenly reversed the legend; thank you for your observation! We will correct it in the Camera-Ready version.

---

### Official Review · Reviewer_ypW5 · 2024-03-08

**Q2-1 Originality-Novelty:** 3
**Q2-2 Correctness-Technical Quality:** 3
**Q2-5 Clarity Of Writing:** 2

**Q10 Ethical Concerns:**

No.

**Q1 Summary And Contributions:**

The proximal causal inference method has recently become a popular approach for overcoming unmeasured confounding in causal analyses. However, proximal causal inference methods rely on strong and untestable methods. In this work, the authors focus on the single proxy case and, assuming the transition matrix p(W | U) is partially observed, propose a sum-of-ratios fractional programming method that partially identifies the causal effect. The authors prove that their method converges to a valid bound for causal queries and provide synthetic and semi-synthetic simulations that prove the effectiveness of their method.

**Q2-3 Extent To Which Claims Are Supported By Evidence:**

3: Good: the main claims are supported by convincing evidence (in the form of adequate experimental evaluation, proofs, (pseudo-)code, references, assumptions).

**Q2-4 Reproducibility:**

2: Fair: key resources (e.g. proofs, code, data) are unavailable but key details (e.g. proof sketches, experimental setup) are sufficiently well-described for an expert to confidently reproduce the main results.

**Q3 Main Strengths:**

Overall, I believe this paper makes an interesting and worthwhile attempt at making proximal causal inference more widely applicable in real-world datasets. As the authors discuss, the completeness assumption in proximal causal inference is opaque, difficult to verify, and often known to be violated simply based on domain knowledge. The authors propose skipping this assumption and settling for estimating bounds (which is what we often care about anyways) for causal estimates with proximal causal inference. I believe the main strengths of this paper are as follows:
1. The authors provide a comprehensive description and analysis of their proposed method – PI-SFP. The authors describe their method, algorithm, and theoretical analysis of their method in clear, separate sections. The theoretical claims are also rigorously supported.
2. The synthetic and non-synthetic experiments show promising results. In the fully synthetic experiments, the authors’ results are very accurate and show little error. Meanwhile, in the non-synthetic experiments based off of previous work, the authors’ results are corroborated by previous estimates and even show tighter bounds.
3. This paper’s main approach of using weaker assumptions in exchange for partial identification can potentially be applied to other causal scenarios, such as the double-proxy control case. Hence, this paper could be a possible stepping stone for further research.

**Q4 Main Weakness:**

I believe the main weaknesses of this paper are as follows:
1. My main critique of this paper is that I do not feel convinced (yet) that the partial observability assumption is one that is realistic in practice. Although the authors do give examples such as the recommender system in the Introduction and the information leakage scenario in their fully synthetic experiments, I believe the authors can expand on these examples more to further convince the reader that it is feasible to have access to lower and upper bounds on p(W | U) in real-world datasets. Although observing the bounds of p(W | U) is a weaker condition than fully observing it as required in the existing literature, it is still a fairly heavy assumption given that the unmeasured confounder U is often completely inaccessible. I believe the authors can address this critique by expanding on the examples already present in their paper or by adding a section dedicated to discussing specific and general scenarios where Assumption 1 is reasonably fulfilled.
2. There are grammar mistakes throughout the paper that make it slightly hard to follow. These grammar mistakes, though, do not affect the validity and impact of the authors’ results. Regrettably, this is a simple weakness to address given the wide availability of grammar check tools on the internet.
3. The code for the authors’ simulations and real-world application experiments are not available. Although the authors give detailed experimental setups and pseudocode for their proposed method, this paper’s reproducibility can be greatly improved by providing the actual code that they used.

**Q5 Detailed Comments To The Authors:**

On the top of page 2 in the sentence, “which is so-called completeness condition”, are you missing a “the” here?

In the same paragraph as the comment above, in the sentence “subject to strict bridge function”, the word “function” here should be plural.

For the first sentence in paragraph 2, “These untestable and impractical constraints…”, the grammar seems a bit off, which makes it a bit harder to understand. Perhaps rephrase it?

In section 4 when the authors discuss the four modules of their algorithm, I think it’s okay just to bold the modules instead of also adding parentheses at the end of the modules. In my personal opinion, I do not think adding parentheses helps make the sections of the algorithm more discernible.

The underline and overline notation can get a bit confusing since it may not always be immediately clear whether a line is an underline for the current line of text or an overline for the next line of text. I encourage the authors to consider another notation, but it may also be fine as it is.

I might have missed this, but, in Algorithm 1, what is the initial value of PI-SFP_{error} before the while loop begins? Did the authors mean to use a do while loop here?

I think section 7 can be moved to right after introducing Assumption 1, or shortly after, for better organization of the paper. It may be better to keep all discussions of Assumption 1 in one place.

In the first sentence of section 8, the “We” should not be capitalized.

On page 5, the second line of equation (5) is past the middle column. The authors should reformat this.

Given the authors' thoughtful responses to my concerns and critiques, I have decided to increase my overall score to a weak accept.

**Q9 Complying With Reviewing Instructions:**

Yes

---

> ### Author Rebuttal · Authors · 2024-04-07
>
> **C1** *My main critique of this paper is that I do not feel convinced (yet) that the partial observability assumption is one that is realistic in practice. Although the authors do give examples such as the recommender system in the Introduction and the information leakage scenario in their fully synthetic experiments, I believe the authors can expand on these examples more to further convince the reader that it is feasible to have access to lower and upper bounds on p(W | U) in real-world datasets. Although observing the bounds of p(W | U) is a weaker condition than fully observing it as required in the existing literature, it is still a fairly heavy assumption given that the unmeasured confounder U is often completely inaccessible. I believe the authors can address this critique by expanding on the examples already present in their paper or by adding a section dedicated to discussing specific and general scenarios where Assumption 1 is reasonably fulfilled.*
>
>
> **Ans** Thank you very much for your attention to the hypothesis. It is wonderful advice to add a justification of assumption.
>
> 1) ``Previous consistency`` Just as our consensus, a discussion has already focused on scenarios in which $P(W\mid U)$ is precisely observable and reversible [1]. If there is no requirement for explicit observability, then it is generally unavoidable to encounter the classic bridge function/completeness assumption (e.g., [2]). Such bridge assumption also implies a requirement for reversibility, which still becomes extremely difficult to satisfy when the information in $W$ is not as rich, namely, ``known to be violated simply based on domain knowledge`` (dear RW ypW5). Hence, compared to these assumptions in classical articles, our hypothesis can now be considered a reasonable and effective relaxation, which is ``is very relevant because this situation is likely to appear in applied research.`` (dear RW SNH5). Moreover, in our appendix, we have also discussed the feasibility of our PI-SFP in the continuous case of confounders.
>
> 2) ``Practicality``
> * **Some general cases (just conduct sampling upon $U$)**: This hypothesis is commonly encountered in real situations, with Kuroki&Judea Pearl[1] (page 4) and Li&Judea Pearl [7] specifically illustrating how to sample $U$ to infer approximate/bounded estimates of $P(W\mid U)$, and mentioning such sampling method has been previously and commonly used, such as fundamental work [3,4,5,6].
> * **Concrete example 1 for $P(W\mid U)$ (recommendation system: W = popularity of U)**: There are also some **more practical** examples in our life. For instance, in the context of recommendation systems, a significant amount of work uses the representation of product-user features as a confounder U. However, this representation often includes sensitive information, leading to incomplete observations of U. Building upon this, the popularity ranking of products in different regions and time periods is publicly available information, which can be used as a proxy variable W for products. By analyzing the different purchasing tendencies of various demographic groups, we can obtain upper and lower bounds estimates for the transition matrix $P(W\mid U)$. In this scenario, firstly, $P(W\mid U)$ is often irreversible because popularity ranking information itself can be seen as an indicator/projection, and the information it carries is not as rich as the product features. Secondly, we typically can only estimate the upper and lower bounds of $P(W\mid U)$ (through methods like Bayesian estimation), as the sampling estimation process for U is likely to be biased.
>
> * **Concrete example 2 for $P(W\mid U)$  (privacy protection: W = de-identified U)**: Our PI-SFP framework is also related to privacy-protecting scenerio. urvey collectors often need to gather some sensitive information U. To obtain more accurate responses and avoid the risk of disclosing personal privacy, they often ask survey respondents to answer some yes-or-no questions using the Randomized Response method. We consider the survey results as W, with the following steps: First, the survey respondent flips a coin (with equal probability of heads or tails), and only they know the result. If it lands heads, they answer the question truthfully; if it lands tails, they flip the coin again (with only them knowing the result); if the second toss is headed, they answer “Yes”; if it is tails, they answer “No”. With this setup, even if we do not know U, we can deduce $P(W)$ and $P(W∣U)=[3/4,1/4;1/4,3/4]$. Of course, this simple setup may still expose other sensitive information, such as the joint distribution of $P(Y,U,X)$, which is something the government would not want to see or make public. Therefore, in practical use, for **highly confidential** information which requires the strongest privacy protection, we tend to develop a more complex/dynamic/irreversible
> $P(W∣U)$ (i.e., privacy-protecting algorithms) and manually set its upper and lower bounds.

---

### Official Review · Reviewer_SNHS · 2024-03-21

**Q2-1 Originality-Novelty:** 3
**Q2-2 Correctness-Technical Quality:** 3
**Q2-5 Clarity Of Writing:** 3

**Q1 Summary And Contributions:**

This is a fantastic paper that contributes to the literature in proximal id. The authors assume the standard proxy id structure, one has $P(W,X,Y)$ and $P(W|U)$, for $X$ being a treatment, $Y$, the outcome, $U$, a confounder, and $W$, a proxy for the confounder. As it is recognized by the authors, the standard conditions for Kuroki and Pearl's derivation, require P(W|U) to be complete. The authors contribute to the literature by obtaining partial id bounds for the case where P(W|U) is incomplete, in particular bounded by $\overline{P(W|U)}$ and $\underline{P(W|U)}$. The approach is sound, the authors transform the problem into a constrained optimization one, where they solve it using a sum-of-ratio fractional programming method. I guess this paper is very relevant, because this situation is likely to appear in applied research.

**Q2-3 Extent To Which Claims Are Supported By Evidence:**

3: Good: the main claims are supported by convincing evidence (in the form of adequate experimental evaluation, proofs, (pseudo-)code, references, assumptions).

**Q2-4 Reproducibility:**

2: Fair: key resources (e.g. proofs, code, data) are unavailable but key details (e.g. proof sketches, experimental setup) are sufficiently well-described for an expert to confidently reproduce the main results.

**Q3 Main Strengths:**

To my knowledge, this paper proposes a new solution. There are indeed contributions in terms of partial id for proximal problems such as Ghassami et al, 2023, however I had never seen a paper targeting incompleteness of P(W|U).

 The paper seems to be organized and well-written, and it was straightforward to follow the main claims and some of the derivations.

**Q4 Main Weakness:**

I couldn't find code in the appendix for replicating the material. I understand that the complexity of the paper might prevent including python code in the appendix, however, I wonder if at least main functions should be included.

**Q5 Detailed Comments To The Authors:**

The paper makes important contribution to the literature in proximal causal inference. Also, there is a second contribution in terms of proposing a new method of optimization, which goes beyond the branch-and-bound method for polynomial programming proposed by Duarte et al. (2023) to solve discrete causal inference problems.

One thing I like to see is  more exploration of cases where P(w1|u) is certain, but other w2,w3,..., wn are not.

**Q9 Complying With Reviewing Instructions:**

Yes

---

> ### Author Rebuttal · Authors · 2024-04-07
>
> **C1** *I couldn't find code in the appendix for replicating the material. I understand that the complexity of the paper might prevent including Python code in the appendix. However, I wonder if at least the main functions should be included.*
>
> We appreciate your request for access to the code accompanying our work. Due to certain objective limitations, such as confidentiality agreements, we are currently unable to make the code publicly available. Rest assured, we are committed to sharing the code post-publication immediately after acceptance, as we value transparency and reproducibility in research practices.
>
> Before acceptance, we will endeavour to provide some useful main functions in the anonymous link (although our pseudocode is already very clear) without violating company protection policies. Please note that these codes are absolutely not the final versions (just for better comprehension of our main text), and the official release after our acceptance will prevail (we will release the final official version in this link <https://anonymous.4open.science/r/Rebuttal_supplement_UAI-541C/README.md> and make it public).
>
>
> **C2** *One thing I like to see is more exploration of cases where P(w1|u) is certain, but other w2,w3,..., wn are not.*
>
> We are delighted that you have posed a rather intriguing question. In layman’s terms, in this scenario, the distribution of PU is subject to only one linear constraint. Indeed, we follow your advice and have made some progress on this issue and would like to share with you:
>
>  1) Theoretically, we prove a new theorem after this paper:
>
>
> We use $\mathcal{P}^L$ to denote the feasible region of $P(U)$ when we only get linear constraints upon it. Then suppose $P(X,Y)>0$, and $ {P}(U)$ with its cardinality $d_u \geq 2$ is within the region $\mathcal{P}^L$ . The tight identification region of the interventional probability $ {P}(y \mid d o(x))$ is given by $ \bigcup\_{P(U) \in \mathcal{P}^L}\left[\mathcal{L B}\_{x, y}^{\text {mul }}( {P}(U)), \mathcal{U B}\_{x, y}^{\text {mul }}( {P}(U))\right]$. Due to the space limitation, we deduce the explanation of such mathematical symbols to the link <https://anonymous.4open.science/r/Rebuttal_supplement_UAI-541C/README.md >.
>
> We take this new result into your setting. Your proposed setting is slightly more complex since it also serves additional linear constraints upon the whole problem via $P(y, w1,x) =\sum_u P(w_1 \mid {u})P(y, {u}, x)$, not only the easier-to-notice constraints $P(w1) =\sum_u P(w_1 \mid {u})P({u})$ as in the above result. Hence, our above result serves as a closed-form valid bound.
>
>
> 2) Practically, we replicate our experiments and replace the previous setting upon transition matrix with the new case $P(w1 | U) = [0.2, 0.1, 0.1, 0.2, 0.2], P(w1) = 0.15$, and $W, U$'s cardinalities are both 5. Equivalently, we choose $\underline{P(W \mid U)} = [P(w1 \mid U), \textbf{0}]$ and  $\overline{P(W \mid U)} = [P(w1 \mid U), \textbf{1}]$ in our PI_SFP (here $\textbf{0}$ & $\textbf{1}$ are corresponding matrices). Then we find that the (lower for instance) PI bound is $0.202$, which is nearly the vanilla bound $0.200$ as in our main text. Such a degeneration result is not surprising since it is consistent with the above theorem.

---

### Official Review · Reviewer_1otP · 2024-03-25

**Q2-1 Originality-Novelty:** 3
**Q2-2 Correctness-Technical Quality:** 2
**Q2-5 Clarity Of Writing:** 2

**Q1 Summary And Contributions:**

In this paper, the authors propose an algorithm to estimate the bounds of P(y|do(x)) in the one-proxy setup by partial observability of the transition matrix P(W|U). Additionally, they provide a theoretical analysis of the tightness of the estimated bounds. To support the results the authors present present experiments based on the synthetic and real data.

**Q2-3 Extent To Which Claims Are Supported By Evidence:**

3: Good: the main claims are supported by convincing evidence (in the form of adequate experimental evaluation, proofs, (pseudo-)code, references, assumptions).

**Q2-4 Reproducibility:**

2: Fair: key resources (e.g. proofs, code, data) are unavailable but key details (e.g. proof sketches, experimental setup) are sufficiently well-described for an expert to confidently reproduce the main results.

**Q3 Main Strengths:**

- The authors considered the problem of bounding the causal effect under the assumption of partial observability of the transition matrix P(W|U), i.e. knowing an upper/lower-bounds on transition matrix P(W|U).  This approach is novel as far as I know and I believe it tackles an important problem.
- The authors provide theoretical guarantees regarding the tightness of the bounds, such as convergence to the true value in the limit

**Q4 Main Weakness:**

- I have concerns regarding the correctness of some statements about the efficiency of the proposed algorithm. Please see the detailed comments.
- An assumption that the cardinality of unobserved variable U and the transition matrix P(W|U) partially observable are strong assumptions
- The experimental results were done on a very simple setup of syntactic data.
- The code for the experiments was provided
- The clarity of writing should be improved.

**Q5 Detailed Comments To The Authors:**

- In supplementary material p.12 the authors claim: "Finally, we prove that our algorithm converges to the global optimal solution at an exponential rate". However, theorem 1 gives a rate O( const^(L_n)), where L_n is an integer between log(n)+1 and n. In case when L_n of the order of log(n) then the rate will be linear but not exponential.
- The bound on the tightness of the estimated causal effect is upper-bounded by const^(L_n)*diameter(S_0)^2. Since this is an upper bound on the probability then we would like to have it smaller than 1 (to be informative). However, the size of the S_0 exponentially grows with the dimension d, and therefore the number of iterations we need to make will grow correspondingly.
- The experiments on synthetic data are performed for the case when d=2, which is a very easy case. I believe that for the synthetic data, the experiments should be done for the bigger values of d. Otherwise, I am concerned whether these experiments can support the main claims of the work.
- Experiments on the real-world application should be described better. In the main paper, the authors present the method to bound the probability P(y|do(x)),  but in the experiments, the target is to estimate a specific expectation. I believe that the setup and the methods used for the real data should be discussed better (at least in supplementary materials). Also, I believe that it is important to specify which methods require additional knowledge of transition matrix P(W|U) and which do not.

**Q9 Complying With Reviewing Instructions:**

Yes

---

> ### Author Rebuttal · Authors · 2024-04-07
>
> Thanks for your advice; here, we will answer questions point by point.
>
> **C1** *I have concerns regarding the correctness of some statements about the efficiency of the proposed algorithm. Please see the detailed comments: In supplementary material p.12 the authors claim:* *Finally, we prove that our algorithm converges to the global optimal solution at an exponential rate". However, theorem 1 gives a rate O( const^(L_n)), where L_n is an integer between log(n)+1 and n. In the case where L_n is the order of log(n), the rate will be linear but not exponential.*
>
> **Ans** We sincerely thank you for your thoughtful feedback. Here, we aim to emphasize that the parameter $L_n$ exhibits an exponential decay trend; therefore, for $n$, our convergence rate lies between linear and exponential. We promise will humbly accept your suggestions and make clarifications in our Camera-Ready version.
>
> Additional warm remark (not crucial, just for a supplement): In fact, $L_n$ falls within the range of $[logn, n]$, and under certain conditions (finite and regularity assumptions, as mentioned on pages 29-30, following Ciarlet [2002], Korotov et al. [2016]), it maintains a linear relationship with $n$. This is an additional conclusion mentioned in the appendix of this article. In fact, in this proof process in the appendix, we also give an answer to an open question/hypothesis: during an infinite number of cuts in the simplicial bi-partition process, under which condition could any point in space only be associated with a finite number of simplicial partitions.
>
>
> **C2**  *The bound on the tightness of the estimated causal effect is upper-bounded by const^(L\_n)diameter(S\_0)^2. Since this is an upper bound on the probability, we would like to have it smaller than 1 (to be informative). However, the size of the S_0 exponentially grows with the dimension d, and therefore, the number of iterations we need to make will grow correspondingly.*
>
> **Ans** We sincerely thank you for your advice. Yes. In order to ensure that our algorithm can output a valid bound, the size of $L_n$ in our algorithm needs to be positively correlated with $d$. We cautiously believe that this is more of a curse of high-value issue itself rather than a problem with algorithm design. We are excited about your suggestion for future directions in the paper: designing faster algorithms to handle cases with large confounder cardinality.
>
>
> Furthermore, our theoretical bound is a rather conservative estimate; from a practical perspective, our bound's convergence rate significantly surpasses the theoretical bound. To demonstrate this, we have included additional experiments for cases with a large confounder cardinality; please see the following.
>
> **C3** *The experiments on synthetic data are performed for the case when d=2, which is a very easy case. I believe that for the synthetic data, the experiments should be done for the bigger values of d. Otherwise, I am concerned whether these experiments can support the main claims of the work.*
>
>
> **Ans** Thank you very much for your insightful suggestion! Following your advice, we conducted multiple experiments with different values of d: keeping the same settings as in the original text (Formulation a.5). Specifically, we chose the same setting of $P(X,Y)$ as in our original paper (A.5), and then take uniform-sampling upon $W$ among its total possible spaces with the same cardinality of $U$. Here, we replace d=2 with $d\in \\{2,4,8,16,32\\}$ (we take the lower bound, for instance, and the upper bound is in the same vein).
>
> ｜$\varepsilon $| $d=2$|$d=4$ | $d=8$|$d=16$| $d=32$|
> |--|--|--|--|--|--|
> |0.1| 0.367(0.55%) | 0.352（0.98%） | 0.302 (1.12%)| 0.291 (1.32%)| 0.288 (1.44%)|
> |0.2| 0.329(0.29%) | 0.298 (0.81%) | 0.277 (1.33%)| 0.235 (1.32%) | 0.213 (1.12%)
> |0.3| 0.291(1.00%) | 0.283 (0.98%) | 0.268 (1.30%) | 0.220 (1.20%) | 0.205 (0.99%)
> |0.4| 0.202(2.44%) | 0.200(2.44%) | 0.200(2.44%) | 0.200(2.44%) | 0.200(2.44%) |
>
> Our PI-SFP methods are also feasible for the multi-cardinality cases of confounders.
>
> **C4** *Experiments on the real-world application should be described better. In the main paper, the authors present the method to bound the probability P(y|do(x)), but in the experiments, the target is to estimate a specific expectation. I believe that the setup and the methods used for the real data should be discussed better (at least in supplementary materials). Also, I believe that it is important to specify which methods require additional knowledge of transition matrix P(W|U) and which do not.*

---

### Meta-Review · Area_Chair_65JW · 2024-04-15

I believe this paper is a very solid contribution to the literature of causal effect estimation with a family of useful assumptions on latent confounding that practitioners are willing to formulate. Reviewers gave numerous suggestions for improvement, and I hope the authors follow them to the letter.